# Communication Efficient Federated Learning over Wireless Channels

## Abstract

Large-scale federated learning (FL) over wireless multiple access channels (MACs) has emerged as a crucial learning paradigm with a wide range of applications. However, its widespread adoption is hindered by several major challenges, including limited bandwidth shared by many edge devices, noisy and erroneous wireless communications, and heterogeneous datasets with different distributions across edge devices. To overcome these fundamental challenges, we propose Federated Proximal Sketching (FPS), tailored towards band-limited wireless channels and handling data heterogeneity across edge devices. FPS uses a count sketch data structure to address the bandwidth bottleneck and enable efficient compression while maintaining accurate estimation of significant coordinates. Additionally, we modify the loss function in FPS such that it is equipped to deal with varying degrees of data heterogeneity. We establish the convergence of the FPS algorithm under mild technical conditions and characterize how the bias induced due to factors like data heterogeneity and noisy wireless channels play a role in the overall result. We complement the proposed theoretical framework with numerical experiments that demonstrate the stability, accuracy, and efficiency of FPS in comparison to state-of-the-art methods on both synthetic and real-world datasets. Overall, our results show that FPS is a promising solution to tackling the above challenges of FL over wireless MACs.

## 1 Introduction

In recent years, federated learning has emerged as an important paradigm for training high-dimensional machine learning models when the training data is distributed across several edge devices. However, when training is carried out over wireless channels in a federated setting, a number of challenges arise, including bandwidth limitations, unreliability and noise in communication channels, and statistical heterogeneity (non-identical distribution) in data across edge devices Kairouz et al. (2021). In what follows, we elaborate on three key challenges. Firstly, with the size of real-world datasets and the machine learning model parameters scaling to the order of millions, communicating model parameters from edge devices to the server and back can become a major bottleneck in model training if not handled efficiently. Needless to say, the transmission of model parameters to the central server over wireless channels is noisy and unreliable in nature. In practice, channel noise is inevitable during the training process and will induce bias in learning the global model parameters. Furthermore, the data collected and stored across edge devices is heterogeneous, which adds an extra layer of complexity due to diversity in local gradient updates. If statistical heterogeneity across edge devices is not handled properly, it can significantly extend the training time and cause the global model to diverge, resulting in poor and unstable performance. Therefore, it is of significant importance to design FL algorithms that are resilient to heterogeneous data distributions and reduce communication costs. While there exists siloed efforts investigating the impacts of the above fundamental challenges separately, we devise a holistic approach - Federated Proximal Sketching (FPS) - that tackles these challenges in an integrated manner.

To address the first key challenge of communication bottleneck, we propose the use of count sketch (CS) Charikar et al. (2002) as an efficient compression operator for model parameters, as illustrated in Figure 1. The CS data structure is not only easy to implement but also comes with strong theoretical

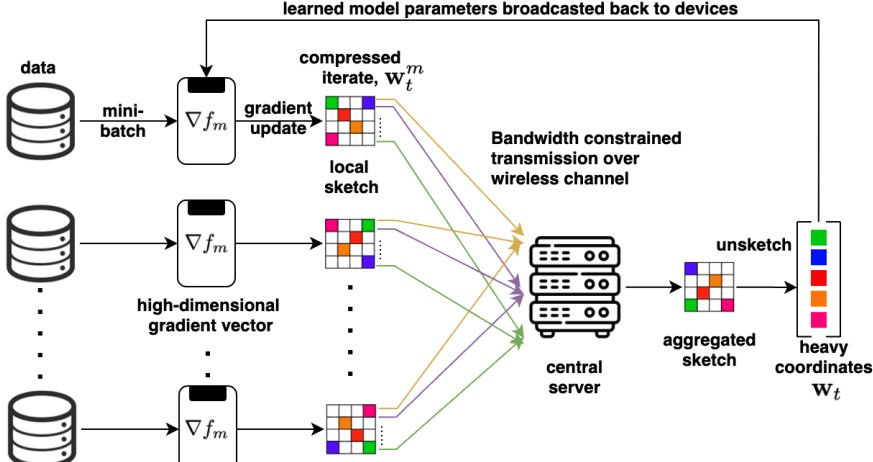

Figure 1: Illustration of Federated Proximal Sketching (FPS) over wireless multi-access channels (MAC).

guarantees on the recovery of significant coordinates or heavy hitters. The CS data structure also enables us to apply the gradient updates easily so that at every time instant, we preserve information about the most important model parameters. With such a compressed representation of the model parameters, over-the-air computing is employed to aggregate local information transmitted by each device. Specifically, over-the-air Abari et al. (2016); Goldenbaum & Stanczak (2013) takes advantage of the superposition property of wireless multiple access channels, thereby scaling signal-to-noise ratio (SNR) well with an increasing number of edge devices.

To tackle the challenges due to data heterogeneity, we employ the proximal gradient method to 'reshape' the loss function by adding a regularization term. The regularization term is carefully designed such that it keeps the learned model parameters from diverging in the presence of data heterogeneity. We use experimental studies to demonstrate empirically that this modification to our loss function helps us reduce the number of communication rounds to the central server while maintaining high accuracy.

The regularization term also helps in curbing the effect of noise due to communication over wireless channels. There is an interesting line of literature highlighted in the Section 2, which studied learning in the presence of noise by using regularization. In addition, we employ the count sketch data structure to produce reliable estimates of the "heavy hitter" (i.e., salient) coordinates. As the count sketch data structure uses multiple hashing functions, the process of sketching and unsketching provides denoising effect, and further the randomized nature of the hash functions produces a noise robust estimates of top-k coordinates. The usage of count sketching in conjunction with regularization forms the core of our strategy to tackle the challenge of FL under noisy wireless channel settings.

The main contributions of this paper can be summarized as follows:

- **Federated Proximal Sketching.** We propose Federated Proximal Sketching (FPS), a novel and robust count-sketch based algorithm for federated learning in noisy wireless environments. FPS is designed to be highly communication-efficient and can effectively handle high-level data heterogeneity across edge devices.
- **Impact of Gradient Estimation Errors.** As the communication of gradient updates over noisy wireless channels may result in bias, we consider a general biased stochastic gradient structure and quantify the impact of gradient estimation errors (including bias). We show that in the presence of biased gradient updates, the FPS algorithm converges with high probability to a neighborhood of the desired global minimum, where the size of the neighborhood hinges upon the bias induced, under mild assumptions. Note that the biased stochastic gradient structure here is more general than the existing line of works on FL Stich et al. (2018); Ivkin et al. (2019); Karimireddy et al. (2020), which do not address the bias in the stochastic gradients, a key aspect in a large number of practical problems.

- **Statistical Heterogeneity.** We theoretically investigate the impact of varying degrees of statistical heterogeneity in data distributed across devices on the convergence. Our study is motivated by Li et al. (2020) to tackle data heterogeneity and extends it to the bandlimited noisy wireless channel setting. A key insight that emerges from our analysis of FPS is that an interplay exists between the degree of data heterogeneity, rate of convergence, and choice of learning rate.
- **Experimental Studies:** We complement our theoretical studies with numerical experiments on both synthetic and real-world datasets. Our experimental results unequivocally demonstrate that FPS exhibits robust performance under noisy and bandlimited channel conditions. To evaluate the performance of our algorithm under varying degrees of class imbalance across edge devices, we investigate different data partitioning strategies. Our results show that, in practice, our algorithm achieves high compression rates on large-scale real-world datasets without significant loss in accuracy across various data distribution strategies. In fact, we observe an improved accuracy of more than $10$-$40\%$ over other competing FL algorithms in highly heterogeneous settings.

## 2 Related Work

Our work looks at federated learning under three key challenges: (1) limited bandwidth across edge devices; (2) noisy wireless MACs; and (3) heterogeneous data distribution across devices. In what follows, we elaborate on different works which have addressed these three challenges.

**Communication Efficient Federated Learning.** Over the years, communication-efficient stochastic gradient descent (SGD) techniques have been developed to reduce transmission costs through various gradient compression methods such as quantization Bernstein et al. (2018); Wu et al. (2018); Alistarh et al. (2017) and sparsification Stich et al. (2018); Aji & Heafield (2017). Sparsification methods like top$-k$ (in absolute value) and random$-k$ have demonstrated theoretical and empirical convergence. However, these approaches rely on the ability to store compression errors locally and reintroduce them in subsequent iterations to ensure convergence Karimireddy et al. (2019). A key limitation of top$-k$ sparsification is the additional communication rounds required between local edge devices to reach consensus on the global top$-k$ (heavy hitters) coordinates at each iteration. In bandlimited settings with a large number of edge devices, this is often impractical. Our work builds on the current research by applying sketching as a compression method in federated learning. Ivkin et al. (2019) proposed a communication-efficient SGD algorithm that uses count-sketch to compress high-dimensional gradient vectors across edge devices. However, their algorithm requires a second round of communication between edge devices and the central server to estimate top$-k$ coordinates, which is often infeasible in practice due to latency and bandwidth limitations. Similarly, Rothchild et al. (2020) introduced FetchSGD, which achieves convergence using sketching without the need for additional communication rounds. However, FetchSGD requires an additional error-accumulation structure at the central server to ensure convergence. Moreover, while the work claims that FetchSGD handles non-IID data distribution effectively, it lacks algorithmic details on addressing heterogeneous data distribution, and it does not provide a comprehensive theoretical or practical analysis in different data heterogeneity scenarios, which we address in our study.

In contrast, our approach is inspired by Aghazadeh et al. (2018b), who employed count-sketch to perform recursive SGD, thus eliminating the need for an additional count-sketch structure to store accumulated errors. In our method, the gradient updates are added to the count-sketch data structure at each time step, where they are aggregated with previous updates, providing a compressed representation of the model parameters. While Aghazadeh et al. (2018b) implemented their approach for a single device, we extend this framework to a federated learning setting over bandlimited noisy wireless channels.

**Federated Learning over Wireless Channels.** In the previous section, we focused on communication-efficient FL under noiseless channels. In practice, transmitting gradient vectors over wireless channels introduces noise and bias. Ang et al. (2020) address this by using regularization-based optimization to mitigate communication-induced bias, inspired by the works of Graves (2011); Goodfellow et al. (2016), where training with noise was approximated through regularization to improve robustness. While many regularizers exist, we opt for $\ell_2$-regularization due to its simplicity and ease of implementation.

Related work also considers mitigating bias from channel noise under power constraints. Zhang et al. (2021); Amiri & Gündüz (2020) propose adaptive power allocation strategies based on channel state information (CSI) and gradient magnitudes to reduce communication error effects (see also Yang et al. (2020); Zhu et al. (2019)). More recently, Wei & Shen (2021) analyzed FedAvg McMahan et al. (2016) under both uplink and downlink noise. While we do not consider power constraints or CSI, our approach can easily be extended to such settings.

**Statistical Heterogeneity across Edge Devices.** One of the core challenges in federated learning, as outlined in Section 1, is the statistical heterogeneity of data across edge devices. Recent developments have led to algorithms like FedProx Li et al. (2020), FedNova Wang et al. (2020b), and SCAFFOLD Karimireddy et al. (2020), all designed to address this challenge by reducing the drift of local iterates from the global iterate maintained at the central server. The theoretical convergence properties of these algorithms have been well-studied under various assumptions that capture the dissimilarity in gradient computation caused by non-IID data distributions Kairouz et al. (2021). We adopt the bounded gradient dissimilarity assumption used in Li et al. (2020), which has been shown to be analogous to other commonly used assumptions such as bounded inter-client variance Li et al. (2021b). However, these algorithms have not been analyzed in the context of band-limited and noisy wireless communication channels. The strategy used in FedProx is particularly relevant to our work, as it addresses statistical heterogeneity by appending a proximal term to the loss function. In later sections, we demonstrate that this proximal term in our algorithm serves two key purposes: reducing the effects of channel noise and facilitating convergence in the presence of statistical heterogeneity.

On the practical side, a recent survey Li et al. (2021a) conducted an extensive experimental study of these state-of-the-art algorithms across various data partitioning strategies and datasets. A data partitioning strategy of particular interest to us is label distribution skewness, where, for example, some hospitals specialize in certain diseases and thus have data specific to those diseases. An extreme form of label distribution skewness occurs when edge devices have access to only a subset of label classes Yu et al. (2020). Another type of label skewness, referred to as class imbalance in modern machine learning literature, was studied in Wang et al. (2020b); Wang et al. (2020a); Yurochkin et al. (2019). We simulate different degrees of statistical heterogeneity by varying the level of class imbalance across edge devices. Our work uniquely intersects the analysis and resolution of the three major challenges in federated learning described above.

## 3 Preliminaries

### 3.1 Federated Learning over Wireless MACs

We consider a federated learning setup where there are $M$ edge devices and a central server. Only a fraction of the dataset $\mathcal{D}$ is available across each of the edge devices such that: $\mathcal{D} = \bigcup_{m=1}^{M} \mathcal{D}_m$. The loss function at an edge device $m$ is defined as: $\ell_m(\mathbf{w}; \mathbf{x}_j, y_i)$, for a data sample $(\mathbf{x}_j, y_j) \in \mathcal{D}_m$. For a mini-batch $\xi^m = \{(\mathbf{x}_j, y_j) : j \in |\xi^m|\}$ sampled at each device $m$, the loss function is denoted as:

$$f_m(\mathbf{w}; \xi^m) \triangleq \frac{\ell_m(\mathbf{w}; \xi^m)}{|\xi^m|}, \tag{1}$$

where, $|\cdot|$ represents cardinality of a set. The objective is to minimize the global loss function given by:

$$f(\mathbf{w}) := \frac{1}{M} \sum_{m=1}^{M} \mathbb{E}_{\xi^m} \left[ f_m(\mathbf{w}; \xi^m) \right] . \tag{2}$$

Here, the expectation is taken with respect to the random process that samples mini-batches at each edge device. The optimization of the loss function in equation 2 is performed iteratively. At time step $t$, we denote the ML model parameters by $\mathbf{w}_t$. At each edge device $m$ and time step $t$, the stochastic gradient is computed using the sampled mini-batch $\xi_t^m$ and expressed as $\mathbf{g}_t^m(\mathbf{w}_t) := \nabla f_m(\mathbf{w}_t; \xi_t^m)$. For simplicity, we denote $\mathbf{g}_t^m(\mathbf{w}_t)$ as $\mathbf{g}_t^m$. These gradients are transmitted over noisy multiple subcarriers via an over-the-air

protocol. We define the aggregated received gradient vector as:

$$\mathbf{g}_t := \frac{1}{M} \sum_{m=1}^{M} \mathbf{g}_t^m + \mathbf{n}_t \,, \tag{3}$$

where $\mathbf{n}_t \in \mathbb{R}^d$ represents the channel noise. The gradient descent update rule at the central server is given by:

$$\mathbf{w}_{t+1} = \mathbf{w}_t - \gamma \, \mathbf{g}_t \,, \tag{4}$$

where $\gamma$ is the fixed learning rate and $\mathbf{w}_{t+1}$ denotes the updated model parameter vector. The updated iterate $\mathbf{w}_{t+1}$ is then broadcasted back to all edge devices. The computation of local stochastic gradients, transmission to the central server, and broadcast of updated iterates is performed recursively.

In general, transmission over wireless channels is noisy, and the number of subcarriers is limited due to bandwidth constraints. Consequently, the received gradient vector $\mathbf{g}_t$ is biased. We discuss the biased structure of stochastic gradients further in Section 5. Next, we elaborate on the count sketch compression operator, which plays a crucial role in bandlimited settings, and discuss its recovery guarantees.

## 3.2 Count Sketch

A count sketch $S$ is a randomized data structure utilizing a $w \times b$ matrix of buckets, where $b$ and $w$ are chosen to achieve specific accuracy guarantees, typically $\mathcal{O}(\log d)$. The algorithm employs $w$ random hash functions $h_j$ for $j \in [w]$ to map the vector's coordinates to buckets, $h_j : \{1, 2, \ldots, d\} \rightarrow \{1, 2 \ldots, b\}$, alongside $w$ random sign functions $s_j$ mapping coordinates to $\{+1, -1\}$.

For a high-dimensional vector $\mathbf{g} \in \mathbb{R}^d$, the count sketch data structure sketches the $i^{th}$ coordinate $\mathbf{g}(i)$ into the cell $S(j, h_j(i))$ by incrementing it with $s_j(i) \, \mathbf{g}(i)$. This process is repeated for each $j \in [w]$ and coordinate $i \in [d]$. In a streaming context, for $T$ updates to $\mathbf{g}$, the count sketch requires $\mathcal{O}\left(\left(k + \frac{||\mathbf{g}_{\text{tail}}||^2}{\varepsilon^2 \, \mathbf{g}(k)}\right) \log d \, T\right)$ memory to provide unbiased estimates of the top-$k$ or heavy hitter (HH) coordinates with high probability:

$$|\hat{\mathbf{g}}(i) - \mathbf{g}(i)| \leq \varepsilon \, ||\mathbf{g}||, \ \forall i \in \text{HH} \,, \tag{5}$$

where HH denotes the indices of the top-$k$ coordinates. All norms $||\cdot||$ refer to the $\ell_2$ norm in Euclidean space unless specified otherwise. Fundamental recovery guarantees for the count sketch can be found in Charikar et al. (2002). It is essential to note that if a vector has too many heavy hitter coordinates, collisions in the count sketch may lead to inaccuracies in the resulting unsketched vector.

## 4 Federated Proximal Sketching

The key steps of the FPS algorithm are outlined in Algorithm 1. Below, we elaborate on the main ideas.

In Steps 1 and 2 of Algorithm 1, the count sketch (CS) data structures at each edge device and the central server are initialized to zero. The size of these CS data structures is determined by the available bandwidth (number of subcarriers, $K$). We proceed with a fixed learning rate at each iteration. The number of local epochs/iterations $E$ to be performed before each global aggregation step is pre-determined. The selection of the number of local epochs is heuristic, and we discuss it in detail in Appendix D.4.

In Steps 5 and 6 of Algorithm 1, the stochastic gradient is computed based on the mini-batch sampled at each edge device. The gradient update vector is formed as $-\gamma \, \mathbf{g}_t^m(\mathbf{w}_t^m)$ and sketched into the CS data structure $S^m$ maintained at that device $m$. Specifically, sketching the gradient update vector into the CS data structure is implemented by the following mathematical operation in Step 6:

$$(-\gamma \, \mathbf{g}_t^m) \rightarrow S^m(\mathbf{w}_t^m) \triangleq S^m(\mathbf{w}_t^m - \gamma \, \mathbf{g}_t^m(\mathbf{w}_t^m)) = S^m(\mathbf{w}_{t+1}^m) \,. \tag{6}$$

This represents the gradient update rule, and implementing it recursively is straightforward due to the linearity property of count sketch (CS) data structures. Notably, this update rule, which compresses the

computed gradient vector into a CS data structure, is reminiscent of the MISSION algorithm presented in Aghazadeh et al. (2018a). However, while MISSION was originally designed for a single device, FPS is a distributed algorithm that executes multiple instances of the MISSION algorithm in parallel. At each iteration in FPS, all edge devices maintain an efficient representation of the learned model parameter vector.

In Steps 8, 9, and 10 of Algorithm 1, the CS data structure at each device is transmitted over noisy wireless multiple access channels based on the frequency of updates pushed to the server. The received sketches are then aggregated, and we perform top-$k$ coordinate extraction to obtain a $k$-sparse vector, $\mathbf{w}_{t+1}$, which is subsequently broadcast back to the edge devices.

Steps 5-10 of Algorithm 1 are executed recursively until convergence is achieved. Given the statistical heterogeneity across devices, aggregating updates after a set number of local updates proves beneficial. In scenarios with high statistical heterogeneity, relying solely on local updates can lead to divergence, as empirically demonstrated by McMahan et al. (2016). To mitigate this issue, we restructure our loss function and outline the advantages of this modification.

**Loss function design.** Our restructuring builds on the work of Li et al. (2020) while adding the benefit of mitigating channel noise effects. The new loss function at each device is given by:

$$f(\mathbf{w}, \mathbf{w}^{gb}) = \ell(\mathbf{w}) + \frac{\mu}{2} \left\| \mathbf{w} - \mathbf{w}^{gb} \right\|^2 , \tag{7}$$

where $\ell(\mathbf{w})$ represents the application-specific loss function, such as cross-entropy loss for binary classification or mean-squared error for linear regression. We denote $\mathbf{w}^{gb}$ as the last aggregated model parameter vector broadcasted by the central server.

With a non-zero proximal parameter $\mu$, this new loss function provides the following benefits; 1) It controls the effects of statistical heterogeneity across devices by preventing the local updates $\mathbf{w}$ from straying too far from the last global update $\mathbf{w}^{gb}$, 2) For an improperly chosen number of local updates $E$, the proximal term minimizes the potential divergence that may result and 3) It imposes a regularization effect on the global iterates, allowing us to bound the $\ell_2$ norm by a positive constant, such that $\|\mathbf{w}^{gb}\|^2 \leq W$.

---

**Algorithm 1** Federated Proximal Sketching (FPS)

---
1: **Inputs:** Number of workers: $M$, mini-batches for each worker $m \in [M]$ at each time step: $\xi_t^m$, local epochs $E$.
2: Initialize individual sketches at each worker $S^m$ with initial model parameters $\mathbf{w}_0^m$: $\mathbf{w}_0 \to S^m = S^m(\mathbf{w}_0)$
3: **for** $t = 1, 2, \ldots, T$ **do**
4:      **for** $m = 1, 2, \ldots, M$ **do**
5:          Compute stochastic gradient using mini-batch $\xi_t^m$: $\mathbf{g}_t^m(\mathbf{w}_t^m)$
6:          Sketch the gradient update vector $(-\gamma \, \mathbf{g}_t^m)$ at each worker: $(-\gamma \, \mathbf{g}_t^m) \to S^m(\mathbf{w}_t^m) = S^m(\mathbf{w}_{t+1}^m)$ and broadcast it to the central server after $E$ local iterations / epochs
7:      **end for**
8:      Receive aggregated sketches at the server: $S_t(\mathbf{w}_{t+1}) = \frac{1}{M} \sum_{m=1}^{M} S^m(\mathbf{w}_{t+1}^m) + n_t$
9:      Unsketch and extract top-k coordinates of parameter vector: $\mathbf{w}_{t+1} = \mathcal{U}_k(S_t(\mathbf{w}_{t+1}))$
10:     Broadcast $k$-sparse parameter vector to all edge devices: $\mathbf{w}_{t+1}^m = \mathbf{w}_{t+1}$
11: **end for**

---

## 5 Convergence Analysis

As is standard, the loss function $f_i$ at each edge device $i$ is assumed to be $L$-smooth non-convex objective function.

**Assumption 1** (*Smoothness*) *A function* $f : \mathbb{R}^d \to \mathbb{R}$ *is* $L-$*smooth if for all* $x, y \in \mathbb{R}^d$, *we have:* $|f(y) - f(x) - \langle \nabla f(x), y - x \rangle| \leq \frac{L}{2} \|y - x\|^2$.

In general, the received aggregate stochastic gradient $\mathbf{g}_t$, is biased, i.e., $(\mathbb{E}[\mathbf{g}_t] \neq \nabla f(\mathbf{w}_t))$, and this can be due to biased stochastic gradient estimation, data heterogeneity across devices and noisy channel conditions

Zhang et al. (2021); Amiri & Gündüz (2020). In what follows, we examine the structure of stochastic gradient vector received at the central server.

**Definition 1** *Given a sequence of iterates $\{\mathbf{w}_t\}_{t=1}^T$, for all $t \in [T]$, the structure of biased stochastic gradient estimator can be written as:*

$$\mathbf{g}_t(\mathbf{w}_t) = \nabla f(\mathbf{w}_t) + \beta_t + \zeta_t, \tag{8}$$

*where $\beta_t$ is the biased estimation error and $\zeta_t$ is the martingale difference noise. The quantities $\beta_t$ and $\zeta_t$ are defined as:*

$$\beta_t := \mathbb{E}_t[\mathbf{g}_t(\mathbf{w}_t)] - \nabla f(\mathbf{w}_t), \quad \zeta_t := \mathbf{g}_t(\mathbf{w}_t) - \mathbb{E}_t[\mathbf{g}_t(\mathbf{w}_t)]. \tag{9}$$

Note that such a structure of the stochastic gradient estimator has been studied in Zhang et al. (2008); Ajalloeian & Stich (2020b). It directly follows from the above definition of bias and martingale difference noise that $\mathbb{E}[\zeta_t] = 0$. Here, the expectation $\mathbb{E}_t[\cdot]$ is taken with respect to $\xi_t$, which represents a realization of a random variable corresponding to the choice of a single training sample in the case of vanilla SGD, or a set of samples in the case of mini-batch SGD, along with the channel noise $n_t$. Furthermore, we assume that the bias and martingale noise terms satisfy the following assumptions.

**Assumption 2** (*Zero mean, $(P_n, \sigma^2)$-bounded noise*) *There exists constants $P_n, \sigma^2 \geq 0$ such that:* $\mathbb{E}_t\left[||\zeta_t||^2\right] \leq P_n ||\nabla f(\mathbf{w}_t)||^2 + \sigma^2$.

**Assumption 3** (*$(P_b, b^2)$-bounded bias*) *There exists constants $P_b \in (0, 1)$ and $b^2 \geq 0$ such that:* $||\beta_t||^2 \leq P_b ||\nabla f(\mathbf{w}_t)||^2 + b^2$.

These assumptions are significantly mild as the second moment bounds of the bias and noise terms scales with true gradient norm and constants $b^2$ and $\sigma^2$ respectively. By setting the tuple $(P_b, P_n, b^2, \sigma^2) = \bar{0}$, we get the special case of unbiased gradient estimators. Convergence for this special case has been well studied in literature.

Next, we turn our attention to the compressibility of gradients. Specifically, we assume that the stochastic gradients are approximately sparse. This is formalized in the following assumption Cai et al. (2022):

**Assumption 4** *The stochastic gradients follow a power law distribution and there exists a $p \in (1, \infty)$ such that $|\mathbf{g}_t(i)| = i^{-p} ||\mathbf{g}_t||$.*

In the Appendix, we show that some of the real-world dataset(s) considered in this paper follow Assumption 4. As the value of $p$ increases we infer that only a small number of coordinates in the vector $\mathbf{g}$ are significant. Therefore by choosing an appropriate size of CS data structure we can ensure efficient compression and strong recovery guarantees of the significant coordinates.

Even though the loss functions across all the devices are same, as the data is distributed in a non-IID manner, due to random sampling of mini-batches across devices there will be dissimilarities in computation of loss functions and their respective gradient estimators. To this end, we define a measure of dissimilarity between gradient estimators across edge devices similar to Li et al. (2020) as follows:

**Definition 2** (*B-local dissimilarity*). *The local functions $f_m$ are $B-$locally dissimilar at $\mathbf{w}$ if $||\mathbb{E}_{\xi_m}[\nabla f_m(\mathbf{w}; \xi_m)]||^2 \leq ||\nabla f(\mathbf{w})||^2 B^2$. We further define $B(\mathbf{w}) = \sqrt{\frac{\mathbb{E}_{\xi_m}[||\nabla f_m(\mathbf{w}; \xi_m)||^2]}{||\nabla f(\mathbf{w})||^2}}$, for $||\nabla f(\mathbf{w})|| \neq 0$.*

Further, we have the following assumption ensuring that the dissimilarity $B(\mathbf{w})$ defined in Definition 2 is uniformly bounded above.

**Assumption 5** *For some $\epsilon > 0$, there exists $B$ such that for all points $\mathbf{w} \in S_\epsilon = \{\mathbf{w} \mid ||\nabla f(\mathbf{w})||^2 > \epsilon\}$, $B(\mathbf{w}) \leq B$.*

If we assume the data is distributed in an IID manner, the same loss function across all devices and the ability to sample an infinitely large sample size, then, $B \to 1$. However, due to different sampling strategies,

in practice, $B > 1$. A larger value of $B$ would imply higher statistical heterogeneity across devices. Other formulations of measuring dissimilarity have been studied in Khaled et al. (2019); Li et al. (2019); Wang et al. (2019).

Let us denote $H = \frac{1}{1 + 2\,B^2 (P_b + P_n)}$. Note that $H \leq 1$. We now define the following quantity $\rho(\gamma)$ as:

$$\rho(\gamma) \triangleq \frac{1 - P_b\,(1 + 2\,H)\,E^2\,B^2}{2} - \gamma\,(2 + 2P_b\,B^2 + (2(L + \mu) + 1)\,P_n\,B^2)\,(1 + 2\,H)\,E^2\,, \tag{10}$$

where, $P_b$, $P_n$, $L$ and $B$ are constants defined earlier; $\mu$ is the proximal parameter of our loss function and $E$ is number of local epochs carried out at each edge device before global aggregation of model parameters at the central server. Let $f(\mathbf{w}^*)$ be the global minimum value of $f$. The range of values of the fixed learning rate $\gamma$ which we consider, satisfies the following conditions: $\rho(\gamma) > 0$ and that is given by:

$$\gamma \leq \frac{1 - 6\,P_b\,E^2\,B^2}{12(1 + P_b\,B^2 + (L + \mu + 1)\,P_n\,B^2)\,E^2}\,. \tag{11}$$

The CS data structure size we consider scales like $\mathcal{O}\left(c\,k\,\log\frac{d\,T}{\delta}\right)$. Here, $c$ is some positive scalar ($c > 1$), $k$ denotes the number of heavy hitter coordinates we are extracting or unsketching from the CS data structure, $d$ is the ambient dimension, $T$ is the number of iterations and $\delta$ is probability of error. We bound the $\ell_2$ norm of the iterates by some positive constant, $||\mathbf{w}||^2 \leq W$. We have the following main theorem on the iterates in the FPS algorithm.

**Theorem 1** *Under Assumptions 1, 2, 3, 4 and 5, the following result holds with probability at least $1 - \delta$:*

$$\frac{1}{T + 1} \sum_{t=0}^{T} \rho(\gamma)\,||\nabla f(\mathbf{w}_t)||^2 \leq \frac{|f(\mathbf{w}_0) - f(\mathbf{w}^*)|}{\gamma\,(T + 1)} + \left(\frac{1}{c} + \frac{(k + 1)^{1-2p} - d^{1-2p}}{2p - 1}\right)\,(L + \mu)^2 W$$

$$+ 2\,E^2\,\left(1 + 2\,P_b\,B^2 + \gamma\,(3 + L + \mu + 2\,P_b\,B^2 + 2\,P_n\,B^2)\right)\,b^2$$

$$+ 2\,E^2\,\left(1 + \gamma\,(L + \mu + 1)(3 + 2\,P_b\,B^2 + 2\,P_n\,B^2)\right)\,\sigma^2\,. \tag{12}$$

**Remarks.** We have a few important observations in order.

- The first term on the right hand side of equation 12 is a scaled version of the term $|f(\mathbf{w}_0) - f(\mathbf{w}^*)|$, and its effect diminishes as $T \to \infty$.
- The second term in equation 12 represents the error in unsketching the top-$k$ coordinates of iterates $\mathbf{w}_t$, viewed as the residual error after extracting top-$k$ coordinates from the CS data structure. With fixed $k$, as the CS size increases, $c$ increases, and $1/c$ becomes smaller. This term also depends on $k$ and $p$: as sketch size grows, more heavy hitters can be extracted, leading to a larger $k$. The value of $p$ is dataset-dependent and reflects how well the input-output relation can be captured by a small feature subset. Fewer significant features result in higher $p$, and vice versa. Thus, as bandwidth increases, the CS size grows, reducing the impact of this term.
- The third and fourth terms in equation 12 capture the effects of bias $\beta_t$ and noise $\zeta_t$. The iterates will likely converge to a neighborhood scaling by $b^2$ and $\sigma^2$. Since $f$ is any non-convex smooth function, multiple contraction regions may exist, each corresponding to a stationary point. These terms grow with increasing local epochs $E$ and data heterogeneity $B$. No fixed $E$ guarantees convergence across varying heterogeneity. As data becomes more heterogeneous, a smaller $E$ reduces the $B^2$ terms, since larger $E$ aggregates bias and noise due to gradient dissimilarity across devices. Thus, more frequent communication with the central server is needed to ensure convergence when heterogeneity increases.
- Analyzing equation 11 shows that the learning rate bound is crucial for convergence. When dissimilarity $B$ is large, a smaller learning rate should be used, as greater dissimilarity increases the likelihood of local models diverging from the global minimum. Thus, a lower learning rate and fewer local epochs help stabilize the algorithm and ensure $\rho(\gamma) > 0$. Moreover, a smaller learning rate reduces the size of the neighborhood scaled by bias and variance in the third and fourth terms of equation 12.

# 6 Experimental Studies

We conduct experiments on synthetic and three real-world datasets with varying model and environmental parameters. Under a bandlimited, noisy wireless channel, we simulate our proposed FPS algorithm and compare it to FetchSGD Rothchild et al. (2020) and bandlimited coordinate descent (BLCD) Zhang et al. (2021). For count sketch-based algorithms (e.g., FetchSGD, FPS), the number of subcarriers dictates the size of the CS structure. For BLCD, random sparsification selects gradient coordinates based on the number of subcarriers. We set the number of edge devices $M = 10$, and model channel noise as zero-mean Gaussian, $\mathcal{N}(0, 1)$. FetchSGD and BLCD perform global aggregation at every epoch, while FPS aggregates every 5 local epochs (chosen heuristically; see Appendix D.4). We use a learning rate of 0.01 in all experiments. To simulate data heterogeneity, we consider four scenarios:

**Scenario 1.** IID distribution with equal class samples across all devices.
**Scenario 2.** Quantity-based label imbalance where each device has samples from only a fixed number of classes (e.g., one class in binary classification).
**Scenario 3.** Distribution-based label imbalance using a Dirichlet distribution $\text{Dir}_M(\alpha)$ to sample class proportions across devices. We set $\alpha = 0.1$ to simulate high label skewness.
**Scenario 4.** Same as Scenario 3, but with $\alpha = 1$ for a more uniform distribution.

For our experimental study, we consider one synthetic dataset and three real-world datasets (KDD12, KDD10, and MNIST). We present the results for the KDD12 and MNIST datasets in the main body of the paper and move the results for the other datasets (synthetic and KDD10) to Appendix D. For all experiments, we report the average accuracy for each data partitioning scenario under both noisy and noise-free conditions. The optimal choice of the proximal parameter, selected from the set $\mu = \{0, 0.01, 0.1, 1\}$ for each scenario, is indicated in the legends below each plot.

## 6.1 KDD12 - Click Prediction

The KDD12 dataset involves a binary classification task where the model must determine if a user will accept {1} or reject {0} an item being recommended, which can include news, games, advertisements, or products. For more details on the dataset, see Juan et al. (2016). The dataset contains $54, 686, 452$ features, and each edge device is allocated $K = 1024$ subcarriers.

In Figure 2, we observe that FPS significantly outperforms FetchSGD and BLCD across all data partitioning strategies and under noisy channel conditions. Additionally, FPS converges more quickly compared to other competing bandlimited algorithms. In Table 1, we report the mean accuracy over 5 trials for various FL algorithms, including FPS, under different degrees of statistical heterogeneity and channel noise conditions.

Specifically, for KDD12, the number of significant coordinates in the gradient update vectors is extremely low compared to the ambient dimension of the dataset (see Appendix E). In this scenario, algorithms like BLCD perform poorly because the probability of randomly selecting significant coordinates when the ambient dimension is high is very low. This poor performance of BLCD is evident in Figure 2. On the other hand, FetchSGD maintains an efficient representation of significant coordinates in the gradient update vectors, so one would expect it to perform well. However, FetchSGD lacks mechanisms to handle noisy wireless channels and data heterogeneity, resulting in subpar performance. The only instances where FetchSGD performs comparably to our algorithm, FPS, are when the data is distributed IID (scenario 1) and when the degree of statistical heterogeneity is low (scenario 4), particularly over noise-free channels (see Table 1). Accuracy plots for FetchSGD, BLCD, and FPS in the noise-free case over varying degrees of data heterogeneity are presented in Figure 7 in Appendix D.

We further evaluate FPS against FedProx (Li et al. (2020)) and top-k FL algorithms, which are not inherently bandlimited. FedProx is a state-of-the-art algorithm designed to learn a global model in the presence of data heterogeneity across edge devices. It communicates the entire gradient update vector to the central server, while the top-k algorithm requires additional communication rounds between edge devices to achieve consensus on global top-k gradient coordinates. Detailed accuracy results are provided in Table 1. In scenarios of extreme heterogeneity (Scenario 2), both FedProx and top-k do not perform well under noisy and noise-free channel conditions. In contrast, under mild statistical heterogeneity (Scenario 4), FedProx

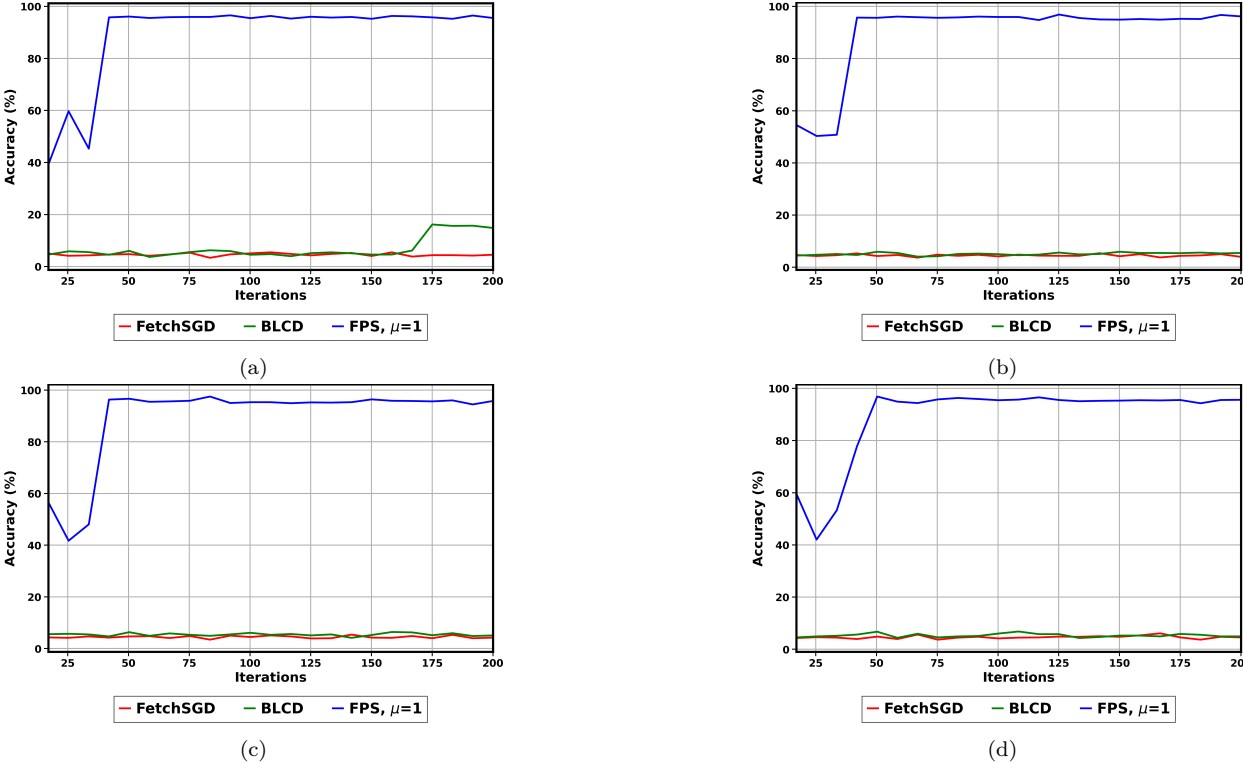

Figure 2: Plotting test accuracy for FPS, BLCD, FetchSGD on KDD12 dataset under noisy channel conditions. The figures correspond to different data partitioning strategies: (a) Scenario 1 (b) Scenario 2 (c) Scenario 3 (d) Scenario 4.

and top-k perform comparably to our FPS algorithm in a noise-free channel setting but struggle in noisy conditions. We hypothesize that the poor performance of FedProx in noisy channels is due to the corruption of approximately sparse gradient update vectors by channel noise. As the less significant coordinates are corrupted, this leads to erroneous gradient updates. While one might argue that scaling the gradient coordinates above the noise floor could resolve this issue, this approach is often infeasible under power constraints.

| Label skewness | Noise $\mathcal{N}(0, \sigma^2)$ | Accuracy (%) | | | | |
|---|---|---|---|---|---|---|
| | | FPS | FetchSGD | BLCD | Top-k | FedProx |
| Scenario 1 | $\sigma = 0$ | $96.44 \pm 0.81$ | $96.48 \pm 1.52$ | $8.51 \pm 2.67$ | $\mathbf{96.64 \pm 0.52}$ | $96.48 \pm 0.81$ |
| | $\sigma = 1$ | $\mathbf{96.56 \pm 1.29}$ | $5.46 \pm 1.33$ | $16.17 \pm 21.23$ | $68.82 \pm 16.66$ | $57.42 \pm 13$ |
| Scenario 2 | $\sigma = 0$ | $\mathbf{97.03 \pm 1.14}$ | $48.12 \pm 1.26$ | $5.93 \pm 1.6$ | $51.09 \pm 2.93$ | $53.20 \pm 6.99$ |
| | $\sigma = 1$ | $\mathbf{96.87 \pm 0.95}$ | $5.39 \pm 0.96$ | $5.93 \pm 1.85$ | $57.57 \pm 24.04$ | $40.93 \pm 11.12$ |
| Scenario 3 | $\sigma = 0$ | $96.64 \pm 0.52$ | $\mathbf{96.79 \pm 0.51}$ | $6.56 \pm 1.38$ | $96.64 \pm 1.22$ | $96.56 \pm 0.67$ |
| | $\sigma = 1$ | $\mathbf{97.5 \pm 0.97}$ | $5.39 \pm 1.24$ | $6.4 \pm 1.27$ | $72.57 \pm 15.3$ | $54.60 \pm 17.26$ |
| Scenario 4 | $\sigma = 0$ | $96.25 \pm 0.76$ | $96.17 \pm 1.08$ | $17.18 \pm 19.55$ | $\mathbf{96.71 \pm 0.46}$ | $96.01 \pm 1.22$ |
| | $\sigma = 1$ | $\mathbf{96.87 \pm 0.95}$ | $6.09 \pm 0.31$ | $6.79 \pm 1.93$ | $66.32 \pm 14.71$ | $46.32 \pm 10.79$ |

Table 1: Test accuracy of different distributed algorithms under varying channel conditions and statistical heterogeneity. For FPS and FedProx, we tune $\mu$ from $\{0, 0.01, 0.1, 1\}$ and report the best accuracy over KDD 12 dataset.

## 6.2 MNIST Dataset

We now consider MNIST dataset to evaluate the performance of our algorithm. For this purpose, we utilize a simple 2-layer neural network with approximately 100,000 parameters (neurons). For communication-

efficient algorithms (FPS, FetchSGD, BLCD), we vary the number of subcarriers as $\{5000, 10000, 20000\}$. The regularization parameter ($\mu$) for the proximal term takes values from the set $\{0, 0.01, 0.1, 1\}$. For count-sketch algorithms (FPS, FetchSGD), the number of top-k heavy hitters extracted varies from $\{2000, 5000, 10000\}$.

In Figure 3, we plot the performance averaged over three trials of different band-limited algorithms, including FPS, under noisy wireless channel conditions across varying degrees of data heterogeneity. A detailed comparison is provided in Table 2, where we also consider other competing federated learning algorithms.

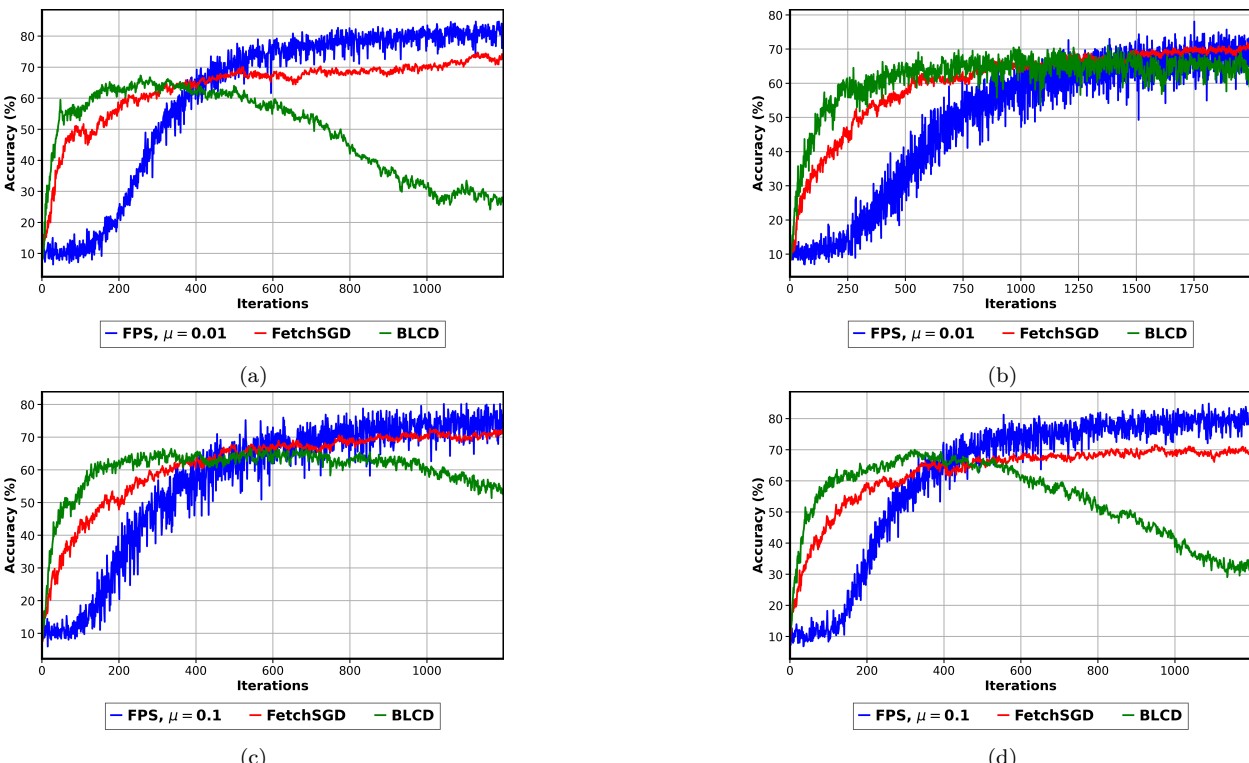

Figure 3: Plotting test accuracy for FPS, BLCD, FetchSGD on MNIST dataset under noisy channel conditions. The figures correspond to different data partitioning strategies (a) Scenario 1 (b) Scenario 2 (c) Scenario 3 (d) Scenario 4.

In the noisy IID case, as depicted in Figure 3a, both FPS and FetchSGD perform well and exhibit comparable accuracies. The slower convergence of FPS can be attributed to the sketching of gradient updates into the count sketch operator, which leads to cancellations and an efficient representation of model parameters at each epoch. By leveraging the existence of a low-dimensional representation of model parameters and consistently sketching gradient vectors in the count-sketch data structure (without re-initializing the CS data structure to zero at the start of each communication round), we can efficiently represent these low-dimensional model parameters after a few epochs. We also observe that the BLCD algorithm exhibits a sudden increase in accuracy followed by a rapid decline in performance. This behavior suggests that while BLCD quickly learns the optimal model parameters, the gradient updates become minimal. Due to noisy wireless channels, these gradient updates are corrupted, resulting in model divergence or a drop in accuracy.

In Scenario 2, for the MNIST dataset, which consists of 10 classes distributed across 10 edge devices such that each edge device has samples corresponding to only one class, as illustrated in Figure 3b, we find that our algorithm maintains robust performance despite extreme data heterogeneity. Interestingly, although BLCD and FetchSGD are not designed to handle data heterogeneity among clients, they still perform reasonably well. Analyzing this unexpected behavior presents an intriguing open question, which we leave for future exploration. In Scenarios 3 and 4, where data heterogeneity is less extreme, we observe in Figures 3c and 3d that FPS continues to outperform or match the performance of FetchSGD. Meanwhile, BLCD does not perform well, mirroring the behavior observed in the noisy IID case.

Referring to Table 2, we conclude that FPS is generally more robust and consistently performs well across different scenarios. Notably, algorithms like FedProx and Top-k, which are not band-limited in nature, only excel in the IID setting or in Scenario 4, where data heterogeneity is minimal. Although our approach shares similarities with the FedProx algorithm in utilizing a proximal term, the results indicate that count-sketch data structures are resilient to additive noise in wireless channels, providing robust estimates of model parameters. We present the plots corresponding to the noise-free case in Appendix D.

| Label | Noise | Accuracy (%) | | | | |
|---|---|---|---|---|---|---|
| skewness | $\mathcal{N}(0, \sigma^2)$ | FPS | FetchSGD | BLCD | Top-k | FedProx |
| Scenario 1 | $\sigma = 0$ | $90.23 \pm 1.79$ | $91.01 \pm 3.01$ | $92.38 \pm 0.57$ | $\mathbf{97.33 \pm 4.8}$ | $91.53 \pm 1.26$ |
| | $\sigma = 0.8$ | $\mathbf{81.31 \pm 1.3}$ | $74.21 \pm 0.87$ | $28.05 \pm 1.29$ | $13.80 \pm 5.61$ | $66.47 \pm 3.22$ |
| Scenario 2 | $\sigma = 0$ | $\mathbf{84.5 \pm 0.82}$ | $13.2 \pm 1.33$ | $8.33 \pm 1.61$ | $10.48 \pm 3.82$ | $8.2 \pm 1.41$ |
| | $\sigma = 0.8$ | $\mathbf{74.54 \pm 1.52}$ | $71.80 \pm 1.79$ | $66.40 \pm 1.26$ | $63.54 \pm 4.76$ | $8.39 \pm 0.36$ |
| Scenario 3 | $\sigma = 0$ | $\mathbf{87.63 \pm 0.57}$ | $65.69 \pm 0.97$ | $56.83 \pm 1.52$ | $64.97 \pm 3.91$ | $28.38 \pm 0.48$ |
| | $\sigma = 0.8$ | $\mathbf{78.38 \pm 0.64}$ | $72.72 \pm 3.53$ | $54.03 \pm 1.11$ | $15.8 \pm 6.09$ | $43.16 \pm 1.7$ |
| Scenario 4 | $\sigma = 0$ | $90.36 \pm 0.54$ | $88.02 \pm 2.5$ | $89.38 \pm 2.5$ | $\mathbf{95.7 \pm 2.76}$ | $89.84 \pm 0.15$ |
| | $\sigma = 0.8$ | $\mathbf{80.27 \pm 0.80}$ | $70.83 \pm 2.48$ | $33.9 \pm 1.7$ | $11.71 \pm 7.3$ | $67.57 \pm 0.9$ |

Table 2: Test accuracy of different distributed algorithms under varying channel conditions and statistical heterogeneity. For FPS and FedProx, we tune $\mu$ from $\{0, 0.01, 0.1, 1\}$ and report the best accuracy over MNIST dataset.

It is important to acknowledge the limitations of our approach. Our algorithm relies on the assumption of an approximately sparse gradient vector and low-dimensional model parameters, which reduces collisions in the count sketch data structure. In cases with dense gradient vectors, compression techniques like count sketch are ineffective, and algorithms such as top-$k$ and BLCD perform better. Each state-of-the-art method excels in specific scenarios, so the choice of FL algorithm depends on the application's requirements.

## 7 Conclusion

We propose Federated Proximal Sketching (FPS), a novel algorithm that learns a global model over bandlimited, noisy wireless channels with data heterogeneity across edge devices. This is the first work to provide both theoretical guarantees and empirical results using sketching as a compression operator under such conditions. We prove that the communication cost to the central server at any round is $\mathcal{O}(\log d)$, significantly lower than the ambient dimension $d$, making FPS efficient for large-scale datasets. Our experiments confirm that FPS's count-sketch compression reduces communication cost with no significant loss in performance.

To simulate data heterogeneity, we employ partitioning strategies based on real-world scenarios. Our results show that adding a proximal term to the loss function stabilizes FPS, preventing divergence under varying heterogeneity and channel noise. We model the impact of data heterogeneity and bias due to noise, and provide convergence results that reveal how parameters like CS size, heterogeneity, bias, and convergence rate interact.

In summary, FPS addresses key challenges in federated learning: data heterogeneity, bandlimited and noisy channels. Our experiments on synthetic and real-world datasets validate the robustness, stability, and superior performance of FPS compared to state-of-the-art algorithms.

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

## Appendix

This appendix is organized as follows: Section A outlines a core element of our paper, the MISSION algorithm. Section B provides the main result for count sketch data structure. Section C provides detailed proofs of main theorem and lemma. Section D discusses the experimental setup and additional empirical results. Section E discusses empirical results supporting the gradient compressibility assumption (Assumption 4) made in the main paper.

## A MISSION Algorithm

The algorithm proposed in Aghazadeh et al. (2018a) is first initialized with a vector $\mathbf{w}_0$ and initialize a count sketch data structure $S$ with zero entries. At iteration $t$, mini-batch stochastic gradient is computed using mini-batch $\xi_t$ and we denoted this as $\mathbf{g}_t$. We form the the gradient update vector by multiplying it with the learning rate: $(-\gamma \mathbf{g}_t)$. We then add the non-zero entries of this computed gradient update vector to the count sketch $S$. Next, MISSION extracts top-$k$ heavy hitters from the sketch, $\mathbf{w}_{t+1}$. The process computation of stochastic gradients and adding it to the sketch is run recursively until the number of iterations desired or until convergence.

---

**Algorithm 2** MISSION

1: Initialize initial vector $\mathbf{w}_0$, Count Sketch $S$ and learning rate $\gamma$
2: **for** $t = 1, 2, \ldots, T$ **do**
3:     Compute stochastic gradient using mini-batch $\xi_t$: $\mathbf{g}_t(\mathbf{w}_t)$
4:     Sketch the local vector $(-\gamma \mathbf{g}_t)$ into $S(\mathbf{w}_t)$: $S(\mathbf{w}_t - \gamma \mathbf{g}_t)$
5:     Unsketch and extract parameter vector: $\mathbf{w}_{t+1} = \mathcal{U}_k(S(\mathbf{w}_{t+1}))$
6: **end for**
7: **Return:** The top-$k$ heavy-hitters of parameter vector $\mathbf{w}$ from the Count-Sketch

---

## B Count Sketch

We now state the main theorem of count sketch data structure.

**Theorem 2 (Count-sketch).** *For a vector $\mathbf{g} \in \mathbb{R}^d$, count sketch recovers the top-k coordinates with error $\pm\varepsilon||\mathbf{g}||_2$ with memory $\mathcal{O}\left(\left(k + \frac{||\mathbf{g}^{tail}||^2}{\varepsilon^2 \mathbf{g}(k)^2}\right) \log \frac{d}{\delta}\right)$; where $||\mathbf{g}^{tail}||^2 = \sum_{i \notin top-k}(\mathbf{g}(i))^2$ and $\mathbf{g}(k)$ is the k-th largest coordinate and this holds with probability at least $1 - \delta$.*

For a detailed proof, we refer to Charikar et al. (2002) .

## C Theoretical Proofs

### C.1 Proof of Lemma 1

Here we state a lemma that upper bounds the residual error after unsketching top-$k$ coordinates of the iterates. This lemma follows directly from the initial recovery guarantees derived in Charikar et al. (2002). We uniformly bound the iterates above by a positive constant $W$ such that: $\mathbb{E}\left[||\mathbf{w}||^2\right] \leq W$. Though this might seem like a bold assumption, we empirically validate that this is true in Section E. We denote the unsketched top-$k$ coordinates of the iterate $\mathbf{w}_t$ as $\tilde{\mathbf{w}}_t$. Here, the subscript $t$ denotes the time index. Under Assumption 4 and the recovery guarantees stated in Theorem 2 we state the following lemma.

**Lemma 1** *If the Count Sketch algorithm recovers the top-k coordinates with error $\varepsilon = \frac{1}{\sqrt{c\,k}}$ and sketch size scaling like $\mathcal{O}\left(c\,k \log \frac{dT}{\delta}\right)$, the following holds for any iterate $\mathbf{w} \in \mathbb{R}^d$ with probability at least $1 - \frac{\delta}{T}$:*

$$\mathbb{E}\left[||\mathbf{w}_t - \tilde{\mathbf{w}}_t||^2\right] \leq \left(\frac{1}{c} + \frac{(k+1)^{1-2p} - d^{1-2p}}{2p - 1}\right) W \tag{13}$$

**Proof:**

$$\mathbb{E}\left[||\mathbf{w}_t - \tilde{\mathbf{w}}_t||^2\right] = \mathbb{E}\left[||\mathbf{w}_t - \mathcal{U}_k(S(\mathbf{w}_t))||^2\right]$$

$$= \mathbb{E}\left[\sum_{i=1}^{k} |\mathbf{w}_t(i) - \tilde{\mathbf{w}}_t(i)|^2 + \sum_{i=k+1}^{d} (\mathbf{w}_t(i))^2\right]$$

$$= \mathbb{E}\left[\varepsilon^2 k ||\mathbf{w}_t||^2 + \sum_{i=k+1}^{d} (\mathbf{w}_t(i))^2\right]$$

$$= \mathbb{E}\left[\varepsilon^2 k ||\mathbf{w}_t||^2 + \sum_{i=k+1}^{d} i^{-2p} \left(\sum_{j=1}^{t} ||-\gamma \mathbf{g}_j||\right)^2\right]$$

$$\leq \mathbb{E}\left[\varepsilon^2 k ||\mathbf{w}_t||^2 + \sum_{i=k+1}^{d} i^{-2p} \left(\left\|\sum_{j=1}^{t} -\gamma \mathbf{g}_j\right\|\right)^2\right]$$

$$= \left(\varepsilon^2 k + \sum_{i=k+1}^{d} i^{-2p}\right) \mathbb{E}\left[||\mathbf{w}_t||^2\right]$$

$$\leq \left(\frac{1}{c} + \frac{(k+1)^{1-2p} - d^{1-2p}}{2p-1}\right) \mathbb{E}\left[||\mathbf{w}_t||^2\right]$$

$$\leq \left(\frac{1}{c} + \frac{(k+1)^{1-2p} - d^{1-2p}}{2p-1}\right) W. \tag{14}$$

∎

Note that, the larger the sketch size gets; the number of coordinates that we can unsketch increases with higher accuracy ($\varepsilon$ decreases).

## C.2 Proof of Lemma 2

**Lemma 2** *For a step size $\gamma \leq \frac{1}{4E(L+\mu)(1+2B^2(P_b+P_n))}$, we can bound the drift for any $e \in \{0, \ldots, E-1\}$ as,*

$$\mathbb{E}\left[||\mathbf{w}_t^e - \mathbf{w}_t||^2\right] \leq 30 E^2 \gamma^2 \left((1 + 2B^2(P_b + P_n))||\nabla f(\mathbf{w}_t)||^2 + b^2 + \sigma^2\right) \tag{15}$$

**Proof:** Now let us concentrate on the term $||\mathbf{w}_t^e - \mathbf{w}_t||^2$, we get:

$$
\begin{aligned}
\mathbb{E}\left[||\mathbf{w}_t^e - \mathbf{w}_t||^2\right] &= \mathbb{E}\left[||\mathbf{w}_t^{e-1} - \gamma\,\mathbf{g}_t^{e-1} - \mathbf{w}_t||^2\right] \\
&= \mathbb{E}\left[||\mathbf{w}_t^{e-1} - \mathbf{w}_t - \gamma\left(\mathbf{g}_t^{e-1} - \nabla f(\mathbf{w}_t^{e-1}) + \nabla f(\mathbf{w}_t^{e-1}) - \nabla f(\mathbf{w}_t) + \nabla f(\mathbf{w}_t)\right)||^2\right] \\
&\leq \left(1 + \frac{1}{2E-1}\right)\mathbb{E}\left[||\mathbf{w}_t^{e-1} - \mathbf{w}_t||^2\right] \\
&\qquad + 2\,E\,\gamma^2\,\mathbb{E}\left[||\nabla f(\mathbf{w}_t^{e-1}) - \mathbf{g}_t^{e-1} + \nabla f(\mathbf{w}_t^{e-1}) - \nabla f(\mathbf{w}_t) + \nabla f(\mathbf{w}_t)||^2\right] \\
&\leq \left(1 + \frac{1}{2E-1}\right)\mathbb{E}\left[||\mathbf{w}_t^{e-1} - \mathbf{w}_t||^2\right] + 6\,E\,\gamma^2\,\mathbb{E}\left[||\nabla f(\mathbf{w}_t^{e-1}) - \mathbf{g}_t^{e-1}||^2\right] \\
&\qquad + 6\,E\,\gamma^2\,\mathbb{E}\left[||\nabla f(\mathbf{w}_t^{e-1}) - \nabla f(\mathbf{w}_t)||^2\right] + 6\,E\,\gamma^2\,||\nabla f(\mathbf{w}_t)||^2 \\
&\leq \left(1 + \frac{1}{2E-1} + 6\,E\,(L+\mu)^2\,\gamma^2\right)\mathbb{E}\left[||\mathbf{w}_t^{e-1} - \mathbf{w}_t||^2\right] \\
&\qquad + 6\,E\,\gamma^2\left(\mathbb{E}\left[||\beta_t^{e-1} + \zeta_t^{e-1}||^2\right] + ||\nabla f(\mathbf{w}_t)||^2\right) \\
&\leq \left(1 + \frac{1}{2E-1} + 6\,E\,(L+\mu)^2\,\gamma^2\right)\mathbb{E}\left[||\mathbf{w}_t^{e-1} - \mathbf{w}_t||^2\right] \\
&\qquad + 6\,E\,\gamma^2\left(B^2\,(P_b + P_n)\,\mathbb{E}\left[||\nabla f(\mathbf{w}_t^{e-1})||^2\right] + b^2 + \sigma^2 + ||\nabla f(\mathbf{w}_t)||^2\right) \\
&\leq \left(1 + \frac{1}{2E-1} + 6\,E\,(L+\mu)^2\,\gamma^2\right)\mathbb{E}\left[||\mathbf{w}_t^{e-1} - \mathbf{w}_t||^2\right] \\
&\quad + 6\,E\,\gamma^2\left(B^2\,(P_b + P_n)\,\mathbb{E}\left[||\nabla f(\mathbf{w}_t^{e-1}) - \nabla f(\mathbf{w}_t) + \nabla f(\mathbf{w}_t)||^2\right] + b^2 + \sigma^2 + ||\nabla f(\mathbf{w}_t)||^2\right) \\
&\leq \left(1 + \frac{1}{2E-1} + 6\,E\,(L+\mu)^2\,\gamma^2\,(1 + 2\,B^2\,(P_b + P_n))\right)\mathbb{E}\left[||\mathbf{w}_t^{e-1} - \mathbf{w}_t||^2\right] \\
&\qquad + 6\,E\,\gamma^2\left((1 + 2\,B^2\,(P_b + P_n))||\nabla f(\mathbf{w}_t)||^2 + b^2 + \sigma^2\right).
\end{aligned}
$$

We assume $\gamma \leq \frac{1}{4\,E\,(L+\mu)\,(1 + 2\,B^2(P_b + P_n))}$ and using this in our analysis so far we get,

$$
\begin{aligned}
\mathbb{E}\left[||\mathbf{w}_t^e - \mathbf{w}_t||^2\right] &\leq \left(1 + \frac{1}{2E-1} + \frac{6}{16\,(1 + 2\,B^2\,(P_b + P_n))\,E}\right)\mathbb{E}\left[||\mathbf{w}_t^{e-1} - \mathbf{w}_t||^2\right] \\
&\qquad + 6\,E\,\gamma^2\left((1 + 2\,B^2\,(P_b + P_n))||\nabla f(\mathbf{w}_t)||^2 + b^2 + \sigma^2\right) \\
&\leq \left(1 + \frac{1}{2E-1} + \frac{1}{2\,E}\right)\mathbb{E}\left[||\mathbf{w}_t^{e-1} - \mathbf{w}_t||^2\right] \\
&\qquad + 6\,E\,\gamma^2\left((1 + 2\,B^2\,(P_b + P_n))||\nabla f(\mathbf{w}_t)||^2 + b^2 + \sigma^2\right) \\
&\leq \left(1 + \frac{1}{E-1}\right)\mathbb{E}\left[||\mathbf{w}_t^{e-1} - \mathbf{w}_t||^2\right] \\
&\qquad + 6\,E\,\gamma^2\left((1 + 2\,B^2\,(P_b + P_n))||\nabla f(\mathbf{w}_t)||^2 + b^2 + \sigma^2\right).
\end{aligned}
$$

Going recursively,

$$
\begin{aligned}
\mathbb{E}\left[||\mathbf{w}_t^e - \mathbf{w}_t||^2\right] &\leq \sum_{e=0}^{E-1}\left(1 + \frac{1}{E-1}\right)^e 6\,E\,\gamma^2\left((1 + 2\,B^2\,(P_b + P_n))||\nabla f(\mathbf{w}_t)||^2 + b^2 + \sigma^2\right) \\
&= (E-1)\left(\left(1 + \frac{1}{E-1}\right)^E - 1\right) 6\,E\,\gamma^2\left((1 + 2\,B^2\,(P_b + P_n))||\nabla f(\mathbf{w}_t)||^2 + b^2 + \sigma^2\right) \\
&\leq 30\,E^2\,\gamma^2\left((1 + 2\,B^2\,(P_b + P_n))||\nabla f(\mathbf{w}_t)||^2 + b^2 + \sigma^2\right) \tag{16}
\end{aligned}
$$

$\blacksquare$

The last inequality follows from the fact that $\left(1 + \frac{1}{E-1}\right)^E \leq 5$ for all $E > 1$. The proof of the above Lemma loosely follows the proof of Lemma 3 in Reddi et al. (2021). Let us now bound the second moment bounds

of computed stochastic gradient, bias and noise terms.

$$\mathbf{g}_t(\mathbf{w}_t) = \sum_{e=0}^{E-1} \mathbf{g}_t(\mathbf{w}_t^e). \tag{17}$$

Keeping the representation simple, we write $\mathbf{g}_t(\mathbf{w}_t^e) = \mathbf{g}_t^e$. Extending this representation, we can expand the computed gradient based on the general structure as, $\mathbf{g}_t^e = \nabla f(\mathbf{w}_t^e) + \beta_t^e + \zeta_t^e$.

$$\mathbb{E}\left[||\mathbf{g}_t||^2\right] = \mathbb{E}\left[\left|\left|\sum_{e=0}^{E-1} \nabla f(\mathbf{w}_t^e) + \beta_t^e + \zeta_t^e\right|\right|^2\right]$$

$$\leq E\left(\sum_e \left((2 + 2P_b B^2 + P_n B^2)||\nabla f(\mathbf{w}_t^e)||^2 + 2b^2 + \sigma^2\right)\right)$$

$$= (2 + 2P_b B^2 + P_n B^2) E\left(\sum_e ||\nabla f(\mathbf{w}_t^e)||^2\right) + 2 E^2 b^2 + E^2 \sigma^2$$

$$= (2 + 2P_b B^2 + P_n B^2) E\left(\sum_e ||\nabla f(\mathbf{w}_t^e) - \nabla f(\mathbf{w}_t) + \nabla f(\mathbf{w}_t)||^2\right) + 2 E^2 b^2 + E^2 \sigma^2$$

$$\leq 2(2 + 2P_b B^2 + P_n B^2) \underbrace{E\left(\sum_e ||\nabla f(\mathbf{w}_t^e) - \nabla f(\mathbf{w}_t)||^2\right)}_{\text{term 1}}$$

$$+ 2(2 + 2P_b B^2 + P_n B^2) E^2 ||\nabla f(\mathbf{w}_t)||^2 + 2 E^2 b^2 + E^2 \sigma^2. \tag{18}$$

Focusing on bounding term 1 in the above equation, we get:

$$E\left(\sum_e \mathbb{E}\left[||\nabla f(\mathbf{w}_t^e) - \nabla f(\mathbf{w}_t)||^2\right]\right) \leq (L + \mu)^2 E \sum_e \mathbb{E}[||\mathbf{w}_t^e - \mathbf{w}_t||^2]. \tag{19}$$

Using the result derived in Lemma 2 we get,

$$E\left(\sum_e \mathbb{E}\left[||\nabla f(\mathbf{w}_t^e) - \nabla f(\mathbf{w}_t)||^2\right]\right) \leq 30 E^4 (L + \mu)^2 \gamma^2 \left((1 + 2 B^2 (P_b + P_n))||\nabla f(\mathbf{w}_t)||^2 + b^2 + \sigma^2\right)$$

$$\leq \frac{2 E^2}{(1 + 2 B^2 (P_b + P_n))^2} \left((1 + 2 B^2 (P_b + P_n))||\nabla f(\mathbf{w}_t)||^2 + b^2 + \sigma^2\right). \tag{20}$$

Now, using equation 20 in equation 18,

$$\mathbb{E}\left[||\mathbf{g}_t||^2\right] \leq 2 E^2 (2 + 2P_b B^2 + P_n B^2) \left(1 + \frac{2}{(1 + 2 B^2 (P_b + P_n))}\right) ||\nabla f(\mathbf{w}_t)||^2$$

$$+ 2\left(1 + \frac{(2 + 2P_b B^2 + P_n B^2)}{(1 + 2 B^2 (P_b + P_n))^2}\right) E^2 b^2 + 2\left(\frac{1}{2} + \frac{(2 + 2P_b B^2 + P_n B^2)}{(1 + 2 B^2 (P_b + P_n))^2}\right) E^2 \sigma^2. \tag{21}$$

Similarly,

$$||\beta_t||^2 = \left|\left|\sum_{e=0}^{E-1} \nabla\beta_t^e\right|\right|^2$$

$$\leq E\left(\sum_e \left(P_b\,B^2||\nabla f(\mathbf{w}_t^e)||^2 + b^2\right)\right)$$

$$= P_b\,B^2\,E\left(\sum_e ||\nabla f(\mathbf{w}_t^e)||^2\right) + E^2\,b^2$$

$$= P_b\,B^2\,E\left(\sum_e ||\nabla f(\mathbf{w}_t^e) - \nabla f(\mathbf{w}_t) + \nabla f(\mathbf{w}_t)||^2\right) + E^2\,b^2$$

$$\leq 2P_b\,B^2\,E\left(\sum_e ||\nabla f(\mathbf{w}_t^e) - \nabla f(\mathbf{w}_t)||^2\right) + 2P_b\,B^2\,E^2\,||\nabla f(\mathbf{w}_t)||^2 + E^2\,b^2\,.$$

Taking expectation on both sides and using the result derived in equation 20 we get,

$$||\beta_t||^2 \leq 2\,P_b\,B^2\,E^2\left(1 + \frac{2}{(1+2\,B^2\,(P_b+P_n))}\right)||\nabla f(\mathbf{w}_t)||^2 + \frac{2\,E^2}{(1+2\,B^2\,(P_b+P_n))^2}\left(\left(\frac{1}{2} + 2\,P_b\,B^2\right)b^2 + \sigma^2\right)\,. \tag{22}$$

Similarly the upper bound on the second moment of noise $\zeta_t$, we have

$$\mathbb{E}\left[||\zeta_t||^2\right] \leq 2\,P_n\,B^2\,E^2\left(1 + \frac{2}{(1+2\,B^2\,(P_b+P_n))}\right)||\nabla f(\mathbf{w}_t)||^2 + \frac{2\,E^2}{(1+2\,B^2\,(P_b+P_n))^2}\left(b^2 + \left(\frac{1}{2} + 2P_n\,B^2\right)\sigma^2\right)\,. \tag{23}$$

### C.3   Proof of Theorem 1

In this section, we begin by defining some quantities and notations. We define the quantity: $\tilde{\mathbf{w}}_{t+1} = \mathcal{U}_k(S(\mathbf{w}_{t+1}))$. Here, $\mathcal{U}_k(S(\cdot))$ represents the unsketching operation. The subscript $k$ denotes the number of top-$k$ coordinates extracted.

As defined in Assumption 1 of the paper, the application specific loss function is $L-$smooth. We denote this application specific loss function as $\ell(\cdot)$. For instance, for a binary classification task, the loss function can be log-loss. Now, our restructured loss function which is formulated by appending a proximal or regularizer term with the leading constant denoted as: $\mu$. This is given by:

$$f(\mathbf{w}, \mathbf{w}^{gb}) = \ell(\mathbf{w}) + \frac{\mu}{2}\left|\left|\mathbf{w} - \mathbf{w}^{gb}\right|\right|^2\,, \tag{24}$$

where, the iterate $\mathbf{w}^{gb}$ as the last aggregated model parameter vector that was broadcasted by the central server. To simplify, we reduce the notation of $f(\mathbf{w}, \mathbf{w}^{gb})$ to $f(\mathbf{w})$. Here, $\mathbf{w}$ is the current iterate at which the function is being evaluated. Appending such a proximal term preserves the smoothness of the function. Therefore, this new restructured loss function $f(\cdot)$ is $(L+\mu)-$smooth.

We assume that $\gamma \leq \frac{1}{2(L+\mu)}$. Given that $f(\cdot)$ is $(L+\mu)-$smooth,we have that:

$$\mathbb{E}_t[f(\tilde{\mathbf{w}}_{t+1})] \leq f(\tilde{\mathbf{w}}_t) + \langle \nabla f(\tilde{\mathbf{w}}_t), \mathbb{E}_t[\tilde{\mathbf{w}}_{t+1} - \tilde{\mathbf{w}}_t] \rangle + \frac{(L+\mu)}{2} \mathbb{E}_t \left[||\tilde{\mathbf{w}}_{t+1} - \tilde{\mathbf{w}}_t||^2\right]$$

$$= f(\tilde{\mathbf{w}}_t) - \langle \nabla f(\tilde{\mathbf{w}}_t), \gamma \mathbb{E}_t[\mathbf{g}_t] \rangle + \frac{(L+\mu)}{2} \mathbb{E}_t \left[||\gamma \mathbf{g}_t||^2\right]$$

$$= f(\tilde{\mathbf{w}}_t) - \gamma \langle \nabla f(\mathbf{w}_t), \mathbb{E}_t[\mathbf{g}_t] \rangle + \langle \nabla f(\mathbf{w}_t) - \nabla f(\tilde{\mathbf{w}}_t), \gamma \mathbb{E}_t[\mathbf{g}_t] \rangle + \frac{(L+\mu)}{2} \gamma^2 \mathbb{E}_t \left[||\mathbf{g}_t||^2\right]$$

$$\overset{(a)}{\leq} f(\tilde{\mathbf{w}}_t) - \gamma \langle \nabla f(\mathbf{w}_t), \nabla f(\mathbf{w}_t) + \beta_t \rangle + \langle \nabla f(\mathbf{w}_t) - \nabla f(\tilde{\mathbf{w}}_t), \gamma \mathbb{E}_t[\mathbf{g}_t] \rangle$$

$$+ \gamma^2 (L+\mu) \left(||\nabla f(\mathbf{w}_t) + \beta_t||^2 + \mathbb{E}_t \left[||\zeta_t||^2\right]\right)$$

$$\overset{(b)}{\leq} f(\tilde{\mathbf{w}}_t) + \frac{\gamma}{2} \left(-2 \langle \nabla f(\mathbf{w}_t), \nabla f(\mathbf{w}_t) + \beta_t \rangle + ||\nabla f(\mathbf{w}_t) + \beta_t||^2\right)$$

$$+ \langle \nabla f(\mathbf{w}_t) - \nabla f(\tilde{\mathbf{w}}_t), \gamma \mathbb{E}_t[\mathbf{g}_t] \rangle + \gamma^2 (L+\mu) \left(\mathbb{E}_t \left[||\zeta_t||^2\right]\right)$$

$$= f(\tilde{\mathbf{w}}_t) + \frac{\gamma}{2} \left(-||\nabla f(\mathbf{w}_t)||^2 + ||\beta_t||^2\right) + \langle \nabla f(\mathbf{w}_t) - \nabla f(\tilde{\mathbf{w}}_t), \gamma \mathbb{E}_t[\mathbf{g}_t] \rangle$$

$$+ \gamma^2 (L+\mu) \left(\mathbb{E}_t \left[||\zeta_t||^2\right]\right), \tag{25}$$

where, inequality $(a)$ is a consequence of using Young's inequality. Inequality $(b)$ is a direct consequence of using the assumption $\gamma \leq \frac{1}{2(L+\mu)}$ from Lemma 2. To keep our analysis visually easy to follow we abbreviate the quantity $\frac{1}{1+2B^2(P_b+P_n)}$ as $H$.

Continuing on with our proof from equation 25 and utilizing the second moment bounds from equation 21 , equation 22 and equation 23 we get:

$$
\begin{aligned}
\mathbb{E}_t[f(\tilde{\mathbf{w}}_{t+1})] \leq{}& f(\tilde{\mathbf{w}}_t) - \left( \frac{\gamma}{2} - \frac{\gamma\,P_b\,(1+2\,H)\,E^2\,B^2}{2} - 2\,P_n\,(L+\mu)\,(1+2\,H)\gamma^2\,E^2\,B^2 \right)\,||\nabla f(\mathbf{w}_t)||^2 \\
&+ 2\,E^2\,H^2\left( \left(\frac{1}{2}+2\,P_b\,B^2\right)\gamma + \gamma^2\,(L+\mu)\right) b^2 + 2\,E^2\,H^2\left(\gamma + \left(\frac{1}{2}+2\,P_n\,B^2\right)\gamma^2\,(L+\mu)\right)\sigma^2 \\
&+ \mathbb{E}_t[\langle (L+\mu)\,(\mathbf{w}_t-\tilde{\mathbf{w}}_t),\gamma\,\mathbf{g}_t\rangle] \\
\overset{(d)}{\leq}{}& f(\tilde{\mathbf{w}}_t) - \left( \frac{\gamma}{2} - \frac{\gamma\,P_b\,(1+2\,H)\,E^2\,B^2}{2} - 2\,P_n\,(L+\mu)\,(1+2\,H)\gamma^2\,E^2\,B^2 \right)\,||\nabla f(\mathbf{w}_t)||^2 \\
&+ 2\,E^2\,H^2\left( \left(\frac{1}{2}+2\,P_b\,B^2\right)\gamma + \gamma^2\,(L+\mu)\right) b^2 + 2\,E^2\,H^2\left(\gamma + \left(\frac{1}{2}+2P_n\,B^2\right)\gamma^2\,(L+\mu)\right)\sigma^2 \\
&+ \frac{(L+\mu)^2}{2}\,\mathbb{E}_t\left[||\mathbf{w}_t-\tilde{\mathbf{w}}_t||^2\right] + \frac{\gamma^2}{2}\,\mathbb{E}_t\left[||\mathbf{g}_t||^2\right] \\
\leq{}& f(\tilde{\mathbf{w}}_t) + \frac{(L+\mu)^2}{2}\,\boxed{\mathbb{E}_t\left[||\mathbf{w}_t-\tilde{\mathbf{w}}_t||^2\right]} \\
&- \left( \frac{\gamma}{2} - \frac{\gamma\,P_b\,(1+2\,H)\,E^2\,B^2}{2} - 2\,P_n\,(L+\mu)\,(1+2\,H)\gamma^2\,E^2\,B^2 - \gamma^2\,E^2\,(2+2P_b\,B^2+P_n\,B^2)\,(1+2\,H) \right)\,||\nabla f(\mathbf{w}_t)||^2 \\
&+ 2\,E^2\,H^2\left( \left(\frac{1}{2}+2\,P_b\,B^2\right)\gamma + \gamma^2\,(L+\mu) + \left(\frac{1}{H^2}+(2+2P_b\,B^2+P_n\,B^2)\right)\gamma^2 \right) b^2 \\
&+ 2\,E^2\,H^2\left(\gamma + \left(\frac{1}{2}+2P_n\,B^2\right)\gamma^2\,(L+\mu) + \left((2+2P_b\,B^2+P_n\,B^2) + \frac{1}{2\,H^2}\right)\gamma^2 \right)\sigma^2\,.
\end{aligned}
$$

$$
\tag{26}
$$

Let us define the quantity:

$$
\begin{aligned}
\rho(\gamma) &= \frac{1-P_b\,(1+2\,H)\,E^2\,B^2}{2} - 2\,P_n\,(L+\mu)\,(1+2\,H)\gamma\,E^2\,B^2 - \gamma\,E^2\,(2+2P_b\,B^2+P_n\,B^2)\,(1+2\,H) \\
&= \frac{1-P_b\,(1+2\,H)\,E^2\,B^2}{2} - \gamma\,(2+2P_b\,B^2+(2(L+\mu)+1)\,P_n\,B^2)\,(1+2\,H)\,E^2\,.
\end{aligned}
\tag{27}
$$

We want the above defined quantity $\rho(\gamma)$ to be greater than zero. This provides us with a bound on the learning rate and it is given by:

$$
\gamma < \frac{1-P_b\,(1+2\,H)\,E^2\,B^2}{2(2+2P_b\,B^2+(2(L+\mu)+1)\,P_n\,B^2)\,(1+2\,H)\,E^2}
$$

Since $H \leq 1$ we get:

$$
\gamma \leq \frac{1-6\,P_b\,E^2\,B^2}{12(1+P_b\,B^2+(L+\mu+1)\,P_n\,B^2)\,E^2}\,.
\tag{28}
$$

Now averaging from 0 to $T$ on both sides and plugging the bound for residual term (highlighted in red in equation 26) by Lemma 1 the following holds with probability $1 - \delta$:

$$\frac{1}{T+1} \sum_{t=0}^{T} \gamma \, \rho(\gamma) \, ||\nabla f(\mathbf{w}_t)||^2 \leq \frac{|f(\mathbf{w}_0) - f(\mathbf{w}^*)|}{(T+1)} + \left( \frac{1}{c} + \frac{(k+1)^{1-2p} - d^{1-2p}}{2p - 1} \right) (L + \mu)^2 W$$

$$+ 2 E^2 H^2 \left( \left( \frac{1}{2} + 2 P_b \, B^2 \right) \gamma + \gamma^2 \, (L + \mu) + \left( \frac{1}{H^2} + (2 + 2 P_b \, B^2 + P_n \, B^2) \right) \gamma^2 \right) b^2$$

$$+ 2 E^2 H^2 \left( \gamma + \left( \frac{1}{2} + 2 P_n \, B^2 \right) \gamma^2 \, (L + \mu) + \left( (2 + 2 P_b \, B^2 + P_n \, B^2) + \frac{1}{2 \, H^2} \right) \gamma^2 \right) \sigma^2 .$$

Then using the fact that $H \leq 1$ and rearranging terms,

$$\frac{1}{T+1} \sum_{t=0}^{T} \rho(\gamma) \, ||\nabla f(\mathbf{w}_t)||^2 \leq \frac{|f(\mathbf{w}_0) - f(\mathbf{w}^*)|}{\gamma \, (T+1)} + \left( \frac{1}{c} + \frac{(k+1)^{1-2p} - d^{1-2p}}{2p - 1} \right) (L + \mu)^2 W$$

$$+ 2 E^2 \left( \left( \frac{1}{2} + 2 P_b \, B^2 \right) + \gamma \, (L + \mu) + \left( 1 + (2 + 2 P_b \, B^2 + P_n \, B^2) \right) \gamma \right) b^2$$

$$+ 2 E^2 \left( 1 + \left( \frac{1}{2} + 2 P_n \, B^2 \right) \gamma \, (L + \mu) + \left( (2 + 2 P_b \, B^2 + P_n \, B^2) + \frac{1}{2} \right) \gamma \right) \sigma^2$$

$$\leq \frac{|f(\mathbf{w}_0) - f(\mathbf{w}^*)|}{\gamma \, (T+1)} + \left( \frac{1}{c} + \frac{(k+1)^{1-2p} - d^{1-2p}}{2p - 1} \right) (L + \mu)^2 W$$

$$+ 2 E^2 \left( 1 + \gamma \, (L + \mu + 1)(3 + 2 \, P_b \, B^2 + 2 \, P_n \, B^2) \right) \sigma^2$$

$$+ 2 E^2 \left( 1 + 2 P_b \, B^2 + \gamma \, (3 + L + \mu + 2 \, P_b \, B^2 + 2 \, P_n \, B^2) \right) b^2 . \tag{29}$$

## D Experimental Details

The primary motivation for selecting the KDD10 and KDD12 datasets is to address the problem of feature selection in machine learning. Feature selection has numerous applications across various fields, including natural language processing, genomics, and chemistry. The goal of feature selection is to identify a small subset of features that best models the relationship between the input data and output. Therefore, efficiently learning a small subset of features in high-dimensional problems requires effective compression techniques. Consequently, the gradient update vectors satisfy the approximately sparse assumption defined in Assumption 4. For the real-world datasets considered in this paper (KDD10 and KDD12), we demonstrate that the computed stochastic gradient vector at each iteration meets the approximately sparse gradient assumption (Assumption 4) detailed in Appendix E.

### D.1 Synthetic Dataset

**Data generation.** For the synthetic regression task in Scenario 1, we generate observations as $\mathbf{y} = \mathbf{X}\mathbf{w} + 0.01\mathbf{n}$, where $\mathbf{w} \in \mathbb{R}^d$ is the parameter vector and $\mathbf{n} \in \mathbb{R}^d$ is additive Gaussian noise with each $n_i \sim \mathcal{N}(0, 1)$. The design matrix $\mathbf{X} \in \mathbb{R}^{N \times d}$ has rows $\mathbf{X}_i \sim \mathcal{N}(\bar{\mathbf{0}}, \Sigma)$, with $\Sigma_{ii} = i^{-p}$ for $i \in [d]$.

In Scenarios 2-4, we generate observations from two distributions: $\mathbf{X}_i \sim \mathcal{N}(\bar{\mathbf{0}}, \Sigma_1)$ and $\mathbf{X}_i \sim \mathcal{N}(\bar{\mathbf{0}}, \Sigma_2)$. Here, $\Sigma_1 = \Sigma$ as in Scenario 1, and $\Sigma_2$ is diagonal with elements $\Sigma_{ii} = j^{-p}$, where $j$ is a random permutation of $\{1, 2, \ldots, d\}$.

**Experimental setup.** Each device is allocated 256 subcarriers. For FPS, the proximal parameter $\mu$ takes values $\{0, 0.001, 0.01, 0.1\}$. We set the dimension $d = 10000$ and power law degree $p = 5$.

Figure 4 shows the average log test loss over 10 trials for FPS, FetchSGD, and BLCD under noisy bandlimited settings. From left to right, the figures correspond to Scenarios 1-4. FPS achieves the lowest test loss across

all scenarios, with BLCD comparable in Scenario 2 and slightly weaker elsewhere. FetchSGD performs poorly in all cases. The optimal $\mu$ for FPS is indicated in the plot legends.

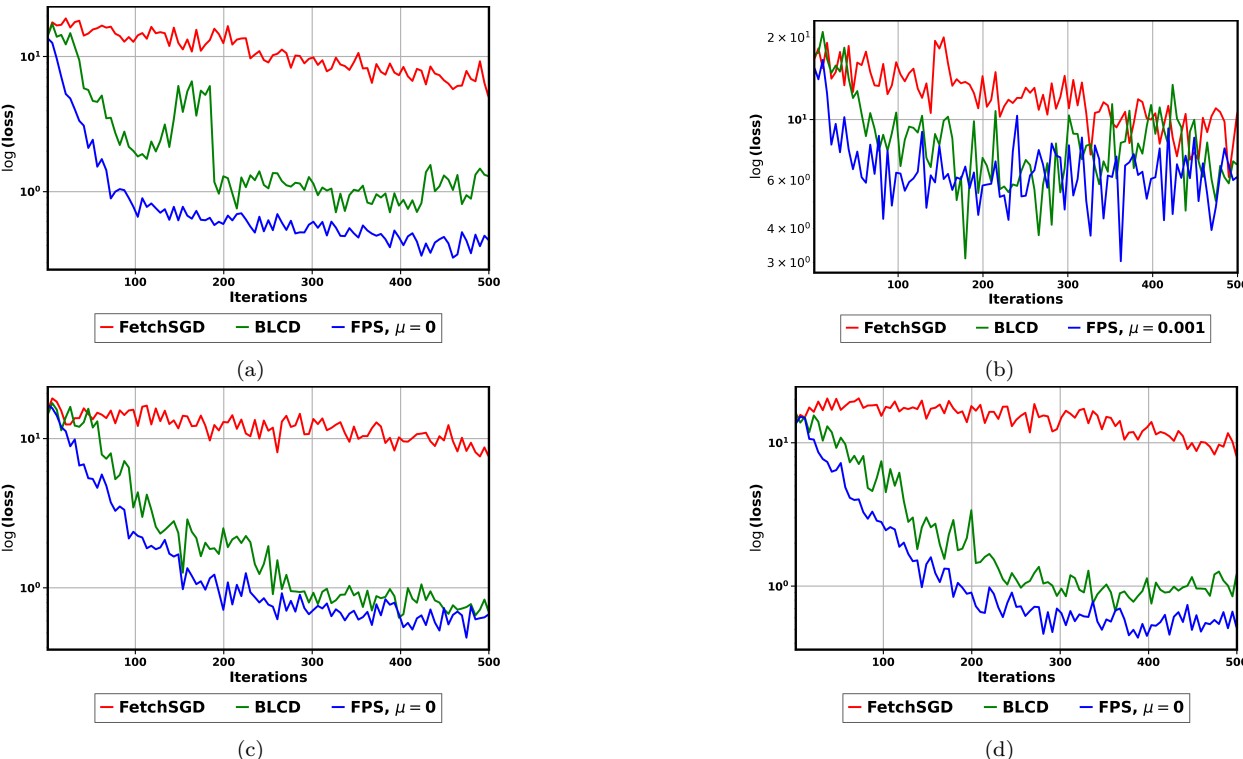

Figure 4: Plotting log of test loss computed for FPS, BLCD, FetchSGD over 5 trials under noisy channel conditions with the gradients following Assumption 4 and power law degree $p = 5$. The figures correspond to different data partitioning strategies: (a) Scenario 1 (b) Scenario 2 (c) Scenario 3 (d) Scenario 4.

### D.1.1 KDD10 Dataset - Predicting Student Performance

The dataset contains $20, 216, 830$ features. For more details, see Yu et al. (2010). Each edge device is allocated $K = 4096$ subcarriers. The number of rows for CS data structure are 5 and the number of columns are 820. The number of top-$k$ significant coordinates that we are extracting are 1000. In Figure 5, we observe that FPS significantly outperforms FetchSGD and BLCD across all data partitioning strategies under bandlimited noisy channel conditions. In Table 3, we report the mean accuracy over 5 trials for various FL algorithms, including FPS, under different degrees of statistical heterogeneity and channel noise conditions.

In Figure 6, we plot the performance of FPS, FetchSGD and BLCD for different data partitioning strategies mentioned in the main paper under noise-free channel conditions on KDD10 dataset. Across all data partitioning scenarios we see that BLCD and FPS perform equally well and better than FetchSGD.

### D.2 KDD12 Dataset

The dataset contains $54, 686, 452$ features. The number of subchannels we consider are 1024. The number of rows for CS data structure are 5 and the number of columns are 204. The number of top-$k$ significant coordinates that we are extracting are 200. In Figure 7, we plot the performance of FPS, FetchSGD and BLCD for different data partitioning strategies mentioned in the main paper under noise-free channel conditions on KDD12 dataset. When the data is distributed in an IID manner (scenario 1), we see that FetchSGD performs slightly better than FPS. In scenario 2 where the data is highly heterogeneous, we see that FPS

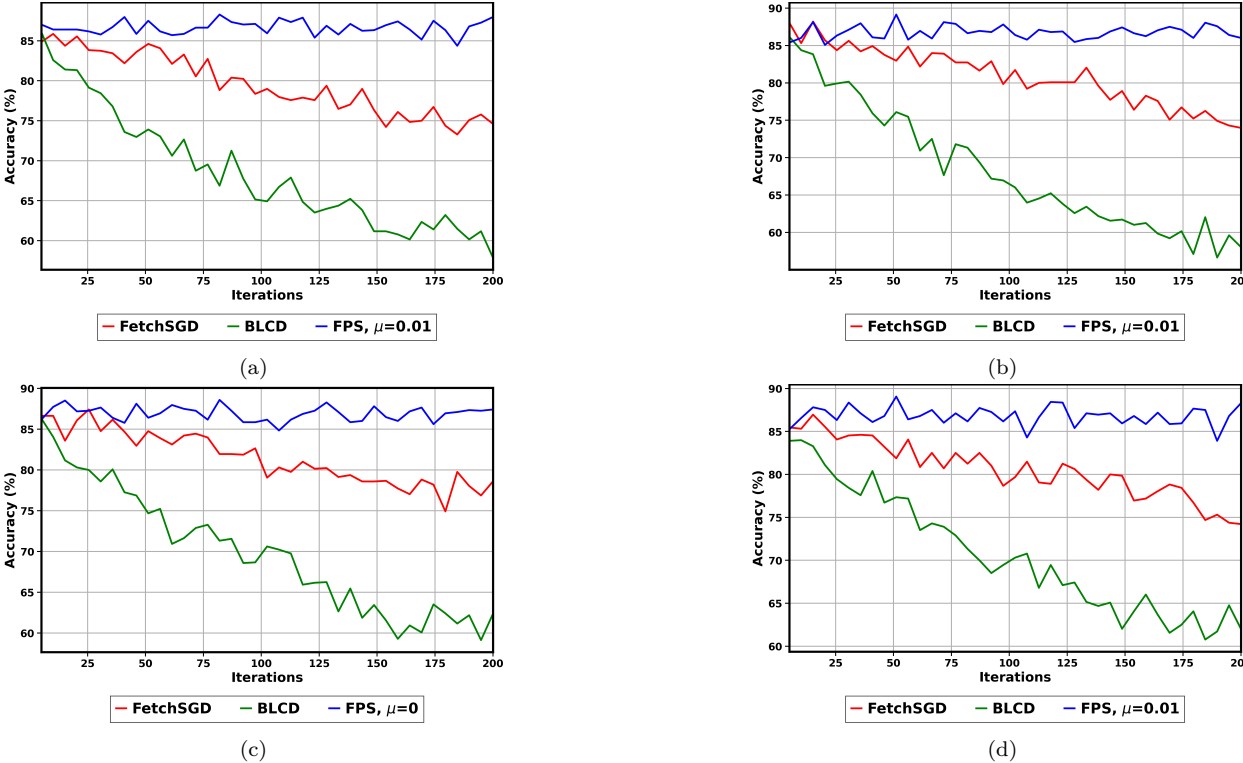

Figure 5: Plotting test accuracy for FPS, BLCD, FetchSGD on KDD10 dataset under noisy channel conditions. The figures correspond to different data partitioning strategies: (a) Scenario 1 (b) Scenario 2 (c) Scenario 3 (d) Scenario 4. We can see that FPS is stable under noisy channel conditions and consistently performs better than other competing bandlimited algorithms.

| Label skewness | Noise $\mathcal{N}(0, \sigma^2)$ | Accuracy (%) | | | | |
|---|---|---|---|---|---|---|
| | | FPS | FetchSGD | BLCD | Top-k | FedProx |
| Scenario 1 | $\sigma = 0$ | $88.04 \pm 1.53$ | $86.64 \pm 1.19$ | $86.79 \pm 2.45$ | $87.10 \pm 1.54$ | $\mathbf{88.12 \pm 2.35}$ |
| | $\sigma = 1$ | $\mathbf{87.96 \pm 1.36}$ | $75.78 \pm 3.84$ | $63.20 \pm 4.15$ | $55.85 \pm 6.15$ | $55.46 \pm 1.69$ |
| Scenario 2 | $\sigma = 0$ | $\mathbf{87.03 \pm 1.66}$ | $54.37 \pm 2.6$ | $72.18 \pm 4.02$ | $54.06 \pm 3.64$ | $55 \pm 1.73$ |
| | $\sigma = 1$ | $\mathbf{88.12 \pm 1.75}$ | $76.25 \pm 3.18$ | $62.03 \pm 2.81$ | $50.07 \pm 3.089$ | $56.71 \pm 3.39$ |
| Scenario 3 | $\sigma = 0$ | $\mathbf{89.68 \pm 1.75}$ | $75.54 \pm 1.68$ | $77.65 \pm 3.21$ | $78.35 \pm 3.11$ | $80.46 \pm 2.26$ |
| | $\sigma = 1$ | $\mathbf{87.42 \pm 2.05}$ | $79.76 \pm 3.40$ | $62.42 \pm 3.37$ | $52.03 \pm 6.01$ | $54.14 \pm 3.86$ |
| Scenario 4 | $\sigma = 0$ | $87.81 \pm 1.96$ | $86.25 \pm 1.44$ | $86.95 \pm 1.72$ | $88.28 \pm 1.71$ | $\mathbf{88.43 \pm 1.12}$ |
| | $\sigma = 1$ | $\mathbf{88.28 \pm 2.06}$ | $76.71 \pm 7.15$ | $64.76 \pm 2.11$ | $59.37 \pm 5.78$ | $56.32 \pm 3.6$ |

Table 3: Test accuracy of different distributed algorithms under varying channel conditions and statistical heterogeneity. For FPS and FedProx, we tune $\mu$ from $\{0, 0.01, 0.1, 1\}$ and report the best accuracy over KDD 10 dataset.

outperforms other competing bandlimited algorithms. In case of scenarios 3 and 4, we see that FPS matches the performance of FetchSGD.

## D.3 MNIST Dataset

For MNIST dataset, we we utilize a simple 2-layer neural network with approximately 100,000 parameters (neurons). For communication-efficient algorithms (FPS, FetchSGD, BLCD), we vary the number of subcarriers as $\{5000, 10000, 20000\}$. The regularization parameter ($\mu$) for the proximal term takes values from the set $\{0, 0.01, 0.1, 1\}$. For count-sketch algorithms (FPS, FetchSGD), the number of top-k heavy hitters extracted varies from $\{2000, 5000, 10000\}$. We report the best accuracy plot over various choices of hyperparameters.

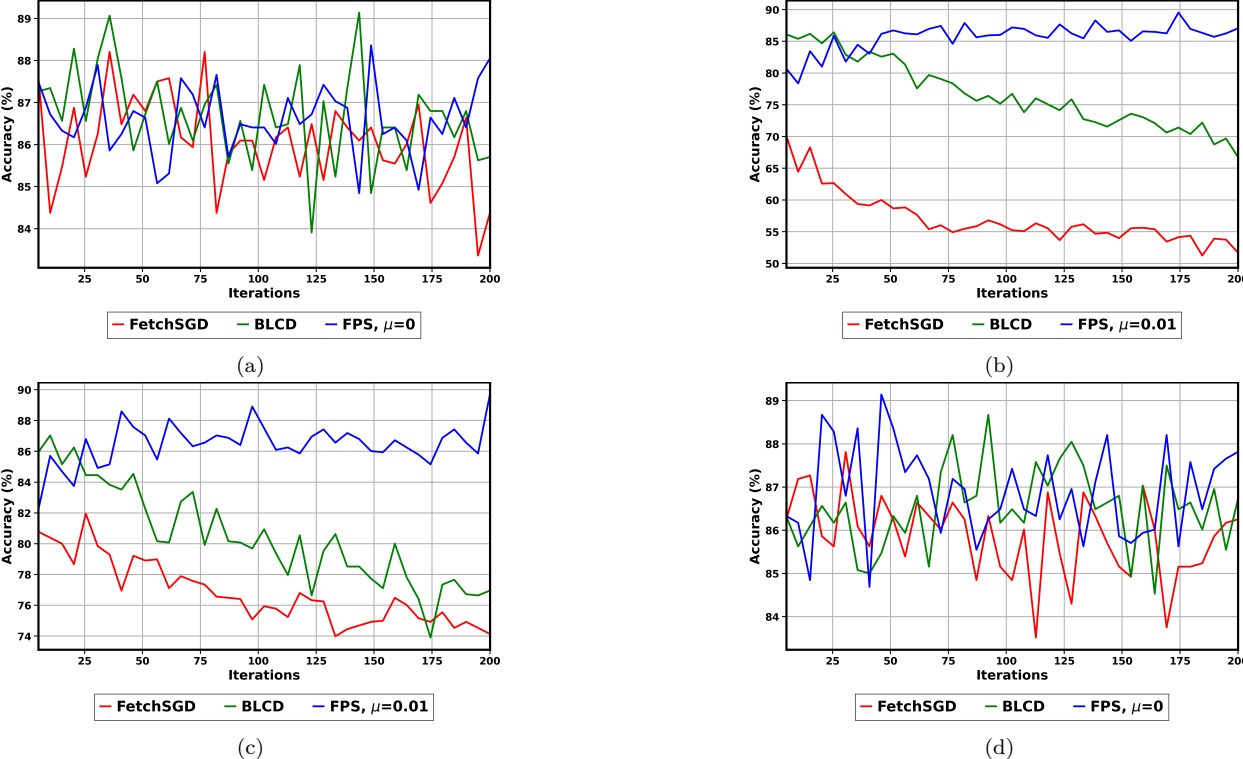

Figure 6: Plotting test accuracy for FPS, BLCD, FetchSGD on KDD10 dataset under noise-free channel conditions. The figures correspond to different data partitioning strategies: (a) Scenario 1 (b) Scenario 2 (c) Scenario 3 (d) Scenario 4.

In Figure 8, we plot the performance of FPS, FetchSGD and BLCD for different data heterogenity scenarios mentioned earlier under noise-free channel conditions on MNIST dataset. Across all scenarios we see that FPS performs well against its band-limited competitors FetchSGD and BLCD.

### D.4 Choosing Hyperparameters

There are two hyperparameters that we consider in the main paper that require further discussion. The first one is the choice of proximal parameter, $\mu$. A large value of $\mu$ will cause the future iterates to be close to the initialization iterate and a low value of $\mu$ may cause the model to diverge. Therefore, the value of proximal parameter must be chosen carefully. In our experiments, we choose the best value of this proximal parameter from a set of values $\{0, 0.01, 0.1, 1\}$. For the two real-world data sets (KDD10 and KDD12) across different data partitioning strategies the best values of $\mu$ are 0.01 and 1 respectively. Note that picking the best value of $\mu$ right away is difficult due to varying statistical heterogeneity and different datasets. An interesting line of work could be finding the ideal choice of proximal parameter automatically. However, another interesting heuristic technique proposed in Li et al. (2020) adaptively tunes $\mu$. For instance, increase $\mu$ when the loss increases and vice versa. We have not examined the effects of such a heuristic in our experiments.

Another hyperparameter that we choose prior to the start of our experiments is number of local updates $E$ performed by each edge device. We choose a uniform $E = 5$ across all edge devices. Choosing a large value of E implies allowing large amounts of work done by edge devices and this can cause the model to diverge when the data is distributed in a non-IID manner. However, to mitigate this we have a proximal term which does not allow the local updates performed by the edge devices in this period to drift far away. However, the choice of an appropriate value of $E$ might be challenging problem in itself as it depends on device constraints and data distribution across all devices.

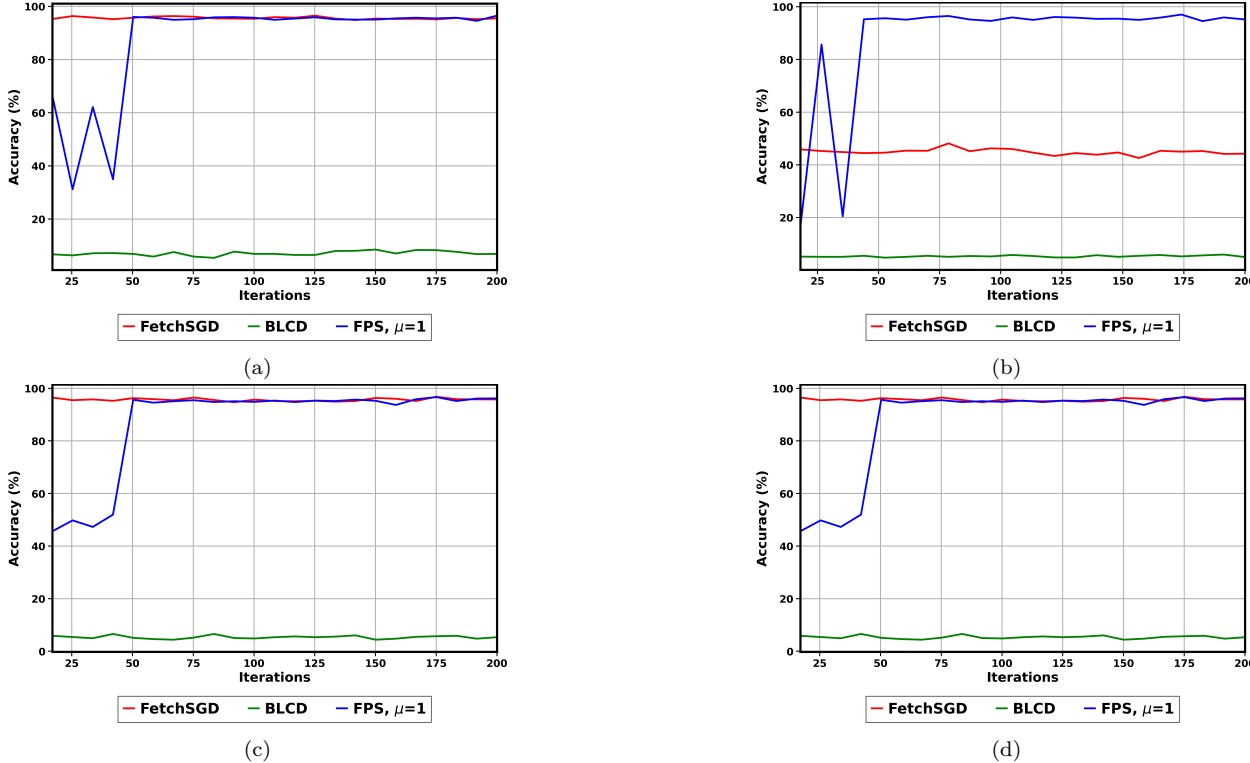

Figure 7: Plotting test accuracy for FPS, BLCD, FetchSGD on KDD12 dataset under noise-free channel conditions. The figures correspond to different data partitioning strategies: (a) Scenario 1 (b) Scenario 2 (c) Scenario 3 (d) Scenario 4.

# E  Gradient Compressibility

The idea that the computed stochastic gradients are compressible or approximately sparse is central to employ efficient compression techniques. In the main paper we formulate mathematically the approximately sparse behaviour of the computed gradients. This needs to be empirically validated as well. We consider the scenario where the data is distributed in an IID manner across devices. We run a federated learning algorithm where there is no bandwidth limitation i.e., high-dimensional gradient vectors are communicated. We consider noise-free channels and the updates are communicated to the central server at every iteration. The loss function has no proximal term appended to it. This naive setup will help us understand the true behaviour of computed stochastic gradients. We run this vanilla FL algorithm for 200 iterations and at the end of it we report $\sim 90\%$ accuracy on both real world datasets (KDD10 and KDD12).

The number of features in the datasets KDD10 and KDD12 are 20,216,830 and 54,686,452 respectively. In Figures 9(a) and 10(a), we plot the absolute value of gradient coordinates computed at a particular edge device for the datasets KDD10 and KDD12 respectively. This plot is captured across three time instants, at iteration 25, 75 and 150. We see that in both figures, the absolute value of coordinates of the local gradient vector sorted in decreasing order are approximately sparse or follow a power law distribution. Similarly in Figures 9(b) and 10(b) we plot the absolute value of coordinates of the aggregated gradient vector received at the central server sorted in decreasing order. This plot is captured across three time instants, at iteration 25, 75 and 150. We observe a similar approximately sparse or power law behaviour for aggregated gradient vectors. If we approximate the number of significant coordinates in computed gradient vectors just by visual inspection of the plots, it is less than 3000. This is far less than the ambient dimension of the datasets we are operating on.

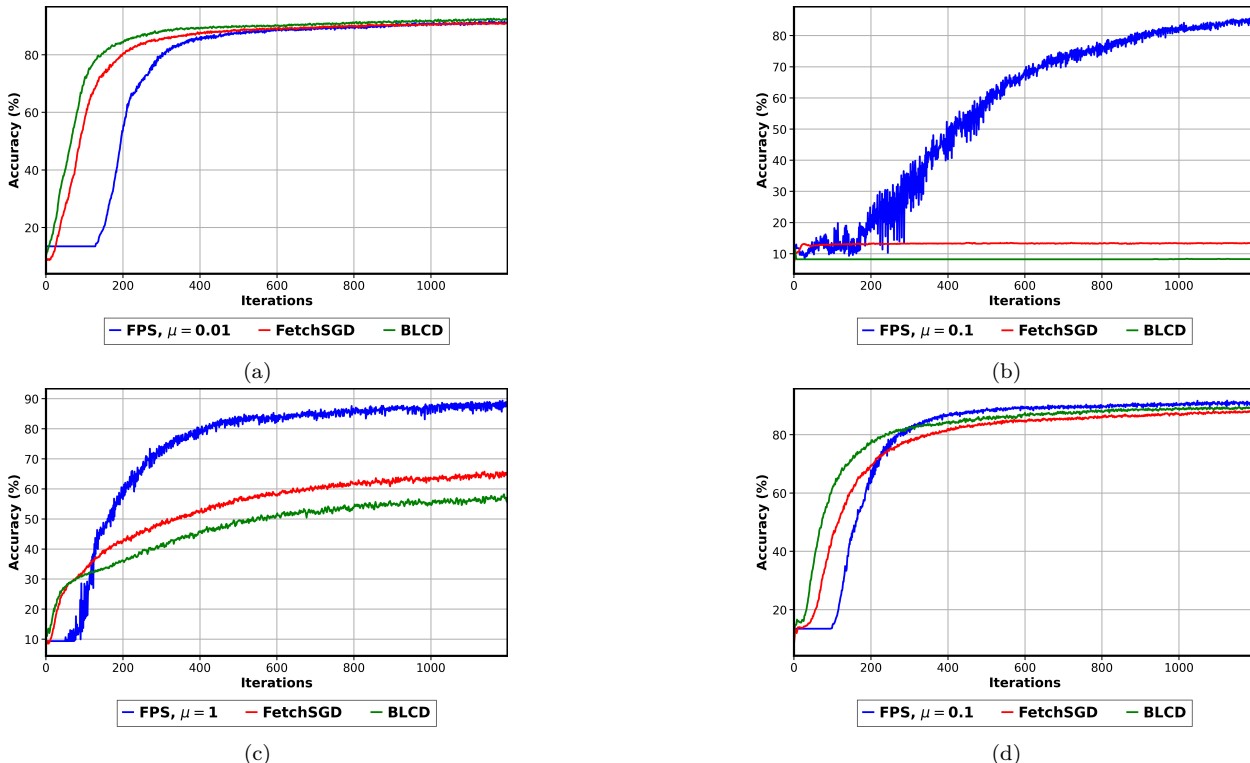

Figure 8: Plotting test accuracy for FPS, BLCD, FetchSGD on MNIST dataset under noise-free channel conditions. The figures correspond to different data partitioning strategies: (a) Scenario 1 (b) Scenario 2 (c) Scenario 3 (d) Scenario 4.

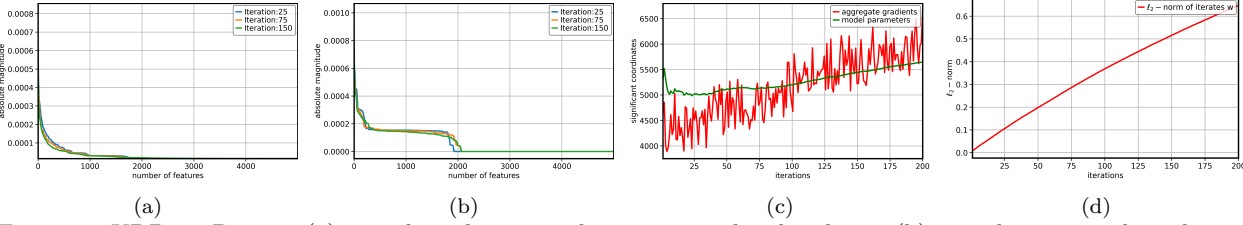

Figure 9: KDD 10 Dataset (a) sorted stochastic gradient at a single edge device (b) sorted aggregated stochastic gradient at the central server (c) significant coordinates of aggregated gradient vector and iterates at the central server (d) $\ell_2-$ norm of iterates.

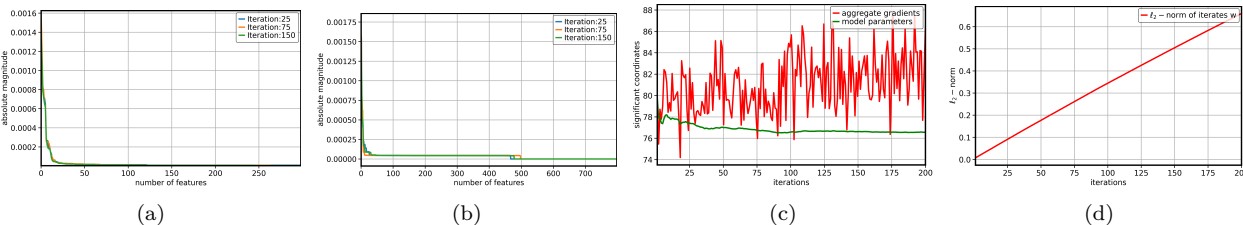

Figure 10: KDD 12 Dataset (a) sorted stochastic gradient at a single edge device (b) sorted aggregated stochastic gradient at the central server (c) significant coordinates of aggregated gradient vector and iterates at the central server (d) $\ell_2-$ norm of iterates.

However, a stronger notion of significant coordinates needs to be used. To this extent we use an alternative measure called *soft* sparsity defined in Lopes (2016):

$$sp(\mathbf{x}) = \frac{||\mathbf{x}||_1^2}{||\mathbf{x}||_2^2} \tag{30}$$

Soft-sparsity represents the number of significant coordinates in a vector. Let $\mathbf{g}$ and $\mathbf{w}$ denote the aggregated gradient and the model parameter vector respectively. For KDD10 dataset, the number of significant coordinates for the aggregated gradient vector $sp(\mathbf{g})$ and the model parameter vector $sp(\mathbf{w})$ are $\sim 5000$, which is much smaller than the ambient dimension. Similarly, for KDD12 dataset, the the number of significant coordinates for the aggregated gradient vector $sp(\mathbf{g})$ are $\sim 85$ and the model parameter vector $sp(\mathbf{w})$ are $\sim 75$. This can be seen in Figures 9(c) and 10(c).

Additionally, we show that the $\ell_2-$norm of the iterates at every iteration received at the central server does not explode and can be uniformly bounded above by a constant. This can be seen in Figures 9(d) and 10(d) for datasets KDD10 and KDD12 respectively.

## F  Dealing with Bias

The vanilla stochastic gradient descent has been well studied in presence of unbiased gradient updates Bottou et al. (2018). Recently, biased gradient updates have been considered in SGD, for instance, in large-scale machine learning systems techniques sparsification, quantization have been used to mitigate the issue of communication bottleneck. Such compression techniques produce biased gradient updates. There is a growing line of work on how different error accumulation and feedback schemes can mitigate the issue of bias and speed up convergence of SGD and distributed learning algorithms Karimireddy et al. (2019); Stich et al. (2018). More recent work on error feedback can be found in Gorbunov et al. (2020); Qian et al. (2021).While this is not the focus of our paper, we are more interested in understanding how bias plays a role in theoretical convergence analysis of SGD. To this extent, we turn towards the body of literature that has dealt with modeling bias into the stochastic gradient structure. Our main motivation to have a more general stochastic gradient structure and mild conditions on bias and noise comes from the work in Ajalloeian & Stich (2020a). Additional works that have considered similar assumptions are Stich (2019); Hu et al. (2021); Bottou (2010) We believe utilizing the assumptions from this line of work into distributed optimization literature (for our paper, FL to be precise) can help us analyze algorithms on a broader scale.

