# OpenReview forum: "Communication Efficient Federated Learning over Wireless Channels"
_TMLR — Rejected by TMLR_

### Review · Reviewer_BskM · 2024-06-13

**Summary Of Contributions:**

The authors present a new algorithm called Federated Proximal Sketching (FPS) to address challenges in noisy wireless environments and data heterogeneity in Federated Learning (FL). They provide a convergence analysis of this algorithm. Additionally, they use a compressed sensing (CS) data structure for model parameter compression and modify the loss function by adding an L2 regularization term to tackle data heterogeneity. They add experiments on the MNIST dataset using neural network and compare the results with other baselines.

**Audience:**

Yes

**Broader Impact Concerns:**

N.A.

**Claims And Evidence:**

Yes

**Requested Changes:**

It would be better to consider more FL algorithms for comparison.

**Strengths And Weaknesses:**

Strengths：
1. The authors provide a theoretical analysis of the convergence of the algorithm .
2. The new algorithm outperforms the other baselines in most cases, and the authors give examples and possible reasons of specific scenarios in which different algorithms may be preferred.

Weaknesses：
Experiments on MNIST dataset show only marginal benefits (or only similar performances) w.r.t. selected baselines.

---

### Review · Reviewer_BScv · 2024-06-21

**Summary Of Contributions:**

This paper studies the issues of limited bandwidth, noisy communications, and heterogeneous datasets for large-scale Federated Learning (FL) over wireless Multiple Access Channels (MACs). The authors propose Federated Proximal Sketching (FPS), which uses a count sketch data structure to efficiently compress data and a regularized loss function to handle data heterogeneity. Under mild conditions (and simple gradient structures; constant learning rate) they study the convergence of their FPS algorithm, which helps understanding the impact of bias from data heterogeneity and noisy channels.

**Audience:**

Yes

**Claims And Evidence:**

Yes

**Requested Changes:**

# Major comments
* The federal learning setup in Section 3.1 is not adequate. When writing the mini-batch \xi^{m} then specify the mini-batch size B, and that it is the input-output pair (x_i,y_i : I\in [B]). Moreover, you state that the loss function convergence to the optimal model parameter vector w^{.}. Please write this as an assumption, and then accordingly also specify/state that the loss function is convex. In addition, on page 8, after eq. (8) you define the global minimum of f with w^{.}.
* You take a fixed learning  rate? Could you add some arguments about why taking this constant when you a large literature on constructing/selecting (adaptive) learning rates for SGD.
* My biggest concern is how the paper addresses biases in mini-batch SGD. Recent work by Godichon-Baggioni et al. (2023) highlights that (i) mini-batch SGD methods can break long- and short-range dependence structures, (ii) biased SGD methods can perform on par with unbiased ones, and (iii) incorporating Polyak-Ruppert averaging can enhance the convergence and robustness. These insights should be considered to deepen the theoretical analysis and improve the robustness of the proposed approach. In particular, they link the mini-batch size, learning rate and the level of bias. Therefore, I would like to know how your setup compares to this? Specifically, what are the (potential) losses of using a constant learning rate (instead of one that follows this size the mini-batches)? And how does Def. 1, Ass. 2, and Ass. 3 compare to their setting, as the noise (Ass. 2) and bias (Ass. 3) are not linked to the mini-batch sizes, which for me is natural.

# Minor comments
* In the experimental studies (Sec. 6) you choose a very small noise, i.e., 0.01. What happens if you choose it larger? And what happens if you take a covariance matrix, \Sigma, to have correlations? C
* Correct citations (into either \citep{} or \citet{}); now, all citations are \citet{}!
* Sometimes the capital letters are for abbreviations and sometimes not. The same applies to section titles.
* Figures and their labels are impossible to read; make labels larger and increase visibility.
* Do a thorough proof-read; there are broken sentence, .e.g., first sentence of Sec. 2.1 and second sentence of Sec. 2.3. In the third point after Thm. 1 you write "equation" twice, and you should switch the bias and noise as they are not third and fourth terms (but fourth and third term). Remove dot in "...Appendix. E.3."
* Revise your appendix structure, such that you start with section "Appendices" and then section "A MISSION Algorithm".

A. Godichon-Baggioni and N. Werge and O. Wintenberger. Learning from time-dependent streaming data with online stochastic algorithms. Transactions on Machine Learning Research (TMLR), 2023.

**Strengths And Weaknesses:**

I want to preface this section by stating that I am not an expert in this area (i.e., federated learning) so I apologize for any inaccuracies.

# Strengths
* Clear theoretical contributions: The theoretical results are well-presented, particularly Theorem 1, which clearly illustrates the trade-off between convergence neighborhood size and data heterogeneity.
* Numerical results: The empirical studies shows that the proposed algorithm can outperform the baselines both on synthetic and real-world datasets, especially under noisy channel conditions.

# Weaknesses
* Theoretical depth: The theoretical analysis, while clear, lacks depth. The assumptions underpinning Theorem 1 are somewhat simplistic and do not fully explore the potential complexities of bias in convergence analysis of SGD.
* Algorithm and dataset diversity: Adding other state-of-the-art algorithms, and including more diverse datasets, beyond MNIST, would enhance the robustness and generalizability of the empirical studies. I naturally become a bit skeptical when seeing a figure as Figure 3, where the baseline does not work. Thus, I start thinking that the baselines may not be the correct baselines to compare to.

---

### Review · Reviewer_oUmf · 2024-09-13

**Summary Of Contributions:**

This paper studies the problem of Federated learning, where individual data silos communicate with a central server via a MAC with additive, potentially biased noise. The algorithm is an evolution of the FetchSGD algorithm that regularizes the drift between local and global iterates and critically, also maintains a sketch of model parameters that are updated with gradients. The authors provide some theoretical results about their algorithm and several experimental settings where their algorithm performs well.

**Audience:**

Yes

**Claims And Evidence:**

Yes

**Requested Changes:**

See weaknesses

* Serious effort to cut down the length of this manuscript
* Address wireless idealizations
* Address power/energy constraint

* You say "In general, transmission over wireless channels is noisy and the number of subcarriers are limited due to bandwidth constraints. As a consequence, the received gradient vector gt is biased." Please elaborate on this. In your experiments, n_t is zero mean gaussian. Is the gradient biased in this section still? In general, I don't understand the full logic of this quoted text. Please provide a rigorous explanation.

**Strengths And Weaknesses:**

Strengths:
* If theorems are correct, (I did not verify all mathematics carefully), then Theorem 1 provides some theoretical results in the presence of bias, which I believe is novel.(*)

* Empirically, this algorithm also seems to perform and outperform the benchmarks.(*)

(*) I'm not entirely up-to-date on this literature, and this is based on my historical reading, and limited literature search I conducted to check.

Weaknesses:

* There in *no reason* this paper needs to exceed the 12 page soft limit. It isn't particularly dense, or introducing an exceedingly complex idea. I would like to see a serious effort to reduce the wordiness and cut down on the length.

* Why do you invoke the term "wireless"? I was quite confused about this at first. I see that you are considering a "MAC with additive noise". This is not really a reasonable model for a wireless network. Please at least mention why you abstract away various important parts of a more reasonable wireless model, though I will not hold you to this point since it seems like many others studying this problem do not do the same. Mention how you ignore things like synchronization, fading, etc.

* How does you model make sense without a power (or energy) constraint? Why can't I just send gradient*100 to avoid noise issues?

Question:

Is there any way to "warm start" you algorithm, by using another algorithm first, this is in ref to Fig. 5. Or in that case would you still need extra iterations for the sketching to catch up?

---

> ### Author Response · Authors · 2024-09-25
> **Addressing wireless idealizations and energy constraints**
>
> **Addressing wireless idealizations such as fading, synchronization, power/energy constraint**
> We acknowledge that our framework simplifies wireless MACs by ignoring additional complexities like fading, synchronization, and power constraints, which have been extensively studied in the literature. However, our primary goal is to highlight the potential of over-the-air (OTA) communication and how its superposition property can be leveraged using the linearity of the count-sketch data structure. Specifically, we focus on how to leverage over-the-air (OTA) communications that naturally enable superposition of client signals using count-sketches. We note that our theory and algorithm naturally allow for the incorporation of more realistic wireless channel properties such as fading and power constraints.
>
> **Implications of Omitting Power Constraints in Gradient Transmission**
> We appreciate the reviewer’s insight regarding power constraints. In this work, we focus on the behavior of the communication protocol in a noise-affected environment. Our primary objective is to examine how the transmitted updates interact with additive noise in over-the-air (OTA) communication and how the resulting bias and variance influence the overall convergence behavior.
>
> The reviewer is correct in noting that scaling the updates with a large positive constant can help mitigate the effects of additive noise. However, as in most realistic scenarios, in the presence of power constraints, such a strategy would be infeasible. In this sense, our work can be viewed as adopting a naive power allocation strategy with equal power distribution across edge devices to expose the dynamics of the learning problem. More sophisticated approaches, such as those in [1,2,3], can readily be applied to our setup to account for channel gains, noise, and optimized power allocation. These approaches would help mitigate the bias and variance introduced by channel noise, while accounting for power constraints. However, the analysis presented here would still be relevant in regimes where one still has to deal with non-trivial amounts of noise. We leave this analysis for future work.[1] J. Zhang, N. Li and M. Dedeoglu, “Federated Learning over Wireless Networks: A Band-limited Coordinated Descent Approach,” _IEEE INFOCOM 2021 - IEEE Conference on Computer Communications_, Vancouver, BC, Canada, 2021.
> [2]X. Cao, G. Zhu, J. Xu and K. Huang, “Optimized Power Control for Over-the-Air Computation in Fading Channels,” in _IEEE Transactions on Wireless Communications_, vol. 19, no. 11, pp. 7498-7513, Nov. 2020.
> [3]M. M. Amiri and D. Gündüz, “Federated Learning Over Wireless Fading Channels,” in _IEEE Transactions on Wireless Communications_, vol. 19, no. 5, pp. 3546-3557, May 2020.

---

> ### Author Response · Authors · 2024-09-25
> **Sources of bias in our setup**
>
> **Bias in our setup**
> There are various sources in our FL setup where bias can arise:
> - **Due to channel conditions**: While additive Gaussian noise is often assumed to have zero mean, non-zero mean noise can occur in real-world communication systems, leading to biased updates. In our experiments, however, we assume zero-mean Gaussian noise, which does not introduce bias, but increases the variance of the estimates.
> - **Due to data heterogeneity across clients**: Another key source of bias in FL systems stems from non-i.i.d. data across clients. When clients have heterogeneous data distributions, the local model updates may become biased toward the local data distribution, causing the global model to diverge from the true underlying distribution. This phenomenon is well-documented in FL literature and can lead to suboptimal convergence.
> - **Due to biased SGD solvers**: The SGD solver used at the client level can also introduce bias, particularly if an improperly tuned optimizer is used. The learning rate, batch size, and other hyperparameters of SGD also affect the quality of local updates. Bias in SGD solvers is further exacerbated when clients have non-i.i.d data resulting in biased gradients being utilized to update model parameters.
> Consequently, Assumptions 2 and 3 in our work account for the bias introduced by all of the above factors. In summary, even when employing zero-mean Gaussian noise in our experiments, the resulting gradient updates remain biased due to other influences.

---

> ### Author Response · Authors · 2024-09-25
> ****Warm start algorithm or how to speed up FPS?****
>
> We hypothesize that the slower convergence of FPS in MNIST case - Figure 4 (Figure 5 in previous version) compared to other baselines can be attributed to the iterative sketching of gradient updates into the count sketch data structure. Specifically, achieving an efficient representation of model parameters requires the cancellation of non-significant coordinates within the count sketch structure, which may take time. A potential solution to accelerate this process is to attenuate the gradient vector on all but its top-$k$ entries at each step (ensuring Assumption 4 - gradient sparsity satisfies with a higher value of $p$).
>
> Our results, based on dataset choice, support this explanation. Datasets like KDD12 and KDD10, which are high-dimensional with few significant features, show faster convergence of FPS compared to other baselines. On such large-scale datasets, gradient updates are sparse (see Appendix E), and non-significant coordinate cancellations occur quickly, leaving only the significant coordinates within the count sketch data structure. In contrast, for MNIST, we suspect that the cancellation of non-significant coordinates is slower, leading to delayed convergence. Exploring the effect of attenuation on convergence speed presents an interesting avenue for future work. Additionally, we will include a brief discussion on this topic in the final version of our paper.

---

> ### Author Response · Authors · 2024-09-25
> **Reduced length of manuscript**
>
> We have made efforts to reduce the length of our manuscript, primarily by making the text more concise. Additionally, we have moved some of the experimental results to the appendix in order to adhere to the soft page limit of 12. We have uploaded a revised copy of our with these changes manuscript.

---

### Author Response · Authors · 2024-08-13
**Performance of FPS against selected baselines and discussion of experimental studies**

## Experiments on datasets where FPS outperforms selected baselines and on MNIST dataset showing only marginal benefits against selected baselines.

1. In the noisy case, we observe that our proposed algorithm performs either better than or on par with other baselines. We refer the reviewers to Figure 8 in the appendix, which shows the performance of our algorithm in the noiseless case; here, it clearly outperforms other baselines significantly. Overall, we see that our proposed approach performs better than competing baselines.

2. One critical point to note is that the effectiveness of our algorithm depends on how compressible the dataset is. Mathematically, in our assumption 4 (approximately sparse gradients) if the value of $p$ is high $\implies$ the number of significant model parameters is low. To this extent, we consider two such datasets  KDD12 and KDD10, which are high dimensional and have only a few significant features. On such large scale datasets:
	1. Algorithms like BLCD, which rely on randomly selecting coordinates at each round, have a high chance of missing important coordinates. However, it is possible that running BLCD for a sufficiently long period will eventually lead to convergence, albeit with a prolonged training duration.
	2. In the case of FetchSGD,  the CS employed to compress gradients is susceptible to corruption due to channel noise, leading to overall poor performance. However, In a noiseless scenario, FetchSGD performs comparably to our approach in most cases. Nevertheless, its inability to handle data heterogeneity causes it to perform poorly in Scenario 2 of Figure 6 (see Appendix, Section D.2).

*Note*: Our proposed approach excels in datasets with a high number of redundant features. Such datasets are commonly found in recommendation systems, genomics, and other problems that require feature selection.

**Summarization of comments on experimental results**
The main advantage of our method lies in applications where the underlying ambient dimension of the dataset is high, and the number of significant features is quite low (datasets like KDD12 and KDD10). We included MNIST in our paper to analyze our proposed method's performance on a widely known dataset.

## Consideration of more FL baselines

The main objective of our paper is to analyze our proposed algorithm (FPS) in a bandlimited noisy channel setting with data heterogeneity across edge devices. To this end, we chose baselines that can operate in such a setting. Firstly, since we operate in a bandlimited regime, we focus on gradient sparsification techniques like random sparsification, which we refer to as bandlimited coordinate descent (BLCD). Secondly, as we use the count-sketch data structure to efficiently compress top-heavy hitter coordinates, we choose FetchSGD as another baseline. FetchSGD uses the count-sketch data structure to efficiently compress the gradient update vectors.

In addition, we included top-$k$ and FedProx as other competing baselines, even though they are not fully suitable for our setting, for the following reasons:

1. **Top-$k$**: This baseline communicates the most significant $k$ coordinates to the central server, making it bandlimited in terms of communication to the central server. However, it requires additional rounds of communication between the edge devices to reach a consensus on which $k$ coordinates to transmit.
2. **FedProx**: As we were motivated by the FedProx-style approach to dealing with statistical heterogeneity across edge devices, we included this as a baseline even though it is not bandlimited.

---

### Author Response · Authors · 2024-08-13
**How our setup compares to Godichon-Baggioni et al. ,2023 and Choice of learning rate.**

## How our work compares to [Godichon-Baggioni et al. ,2023](https://arxiv.org/pdf/2205.12549) .

1. **Problem setup**: One of the main differences between [Godichon-Baggioni et al. ,2023](https://arxiv.org/pdf/2205.12549) and our work is the streaming setting. In the streaming setting, samples drawn at two different time instances do not adhere to the i.i.d assumption, and subsequent gradient estimates will have some form of dependence (short-term or long-term). In our work, the data sample at each client is drawn i.i.d from an unknown distribution.   Therefore, the notion of time-varying dependencies between samples does not apply in our setup.
2. **Biased gradients**: Bias in our setting arises due to the following factors:
	1. The samples at each edge device are not drawn from the same probability distribution, therefore, when computing mini-batch gradient estimates, it leads to biased gradients. Note that this is different from the setting that  [Godichon-Baggioni et al. ,2023](https://arxiv.org/pdf/2205.12549) state where bias arises due to time dependence between samples.
	2. Secondly, bias arises due to channel noise corrupting the gradient updates.

	Mathematically, it is instructive to draw parallels between how bias is assumed to scale in our work vs in [Godichon-Baggioni et al. ,2023](https://arxiv.org/pdf/2205.12549). Recall our assumption on bias (Assumption 3):
	$$
	\begin{align}
	\|\mathbb{E}[\mathbf{g}_t (\mathbf{w}_t)] - \nabla f(\mathbf{w}_t)\|^2 = \|\, \beta_t \|^2  &\leq P_b\, \| \nabla f(\mathbf{w}_t)\|^2 + b^2,
	\end{align}
	$$
	where, $\mathbf{g}_t(\mathbf{w}_t)$ is the biased gradient estimate evaluated at model parameter $\mathbf{w}_t$, $\|\nabla f(\mathbf{w}_t) \|$ is the true gradient estimate. $P_b$ and $b^2$ are some positive constants.

	Now let us state Assumption 3-p from [Godichon-Baggioni et al. ,2023](https://arxiv.org/pdf/2205.12549) for $p = 2$:
$$
	\mathbb{E} [ || \mathbb{E} [ \mathbf{g}_t (\mathbf{w}_t) ] - \nabla f(\mathbf{w}_t) ||^2 \leq v_t² (D_v^2\mathbb{E}\left[ || \mathbf{w}_t - \mathbf{w}^* || ² \right] + B_v^2)
	$$
	We now show that our Assumption 3 (our work) is a version of Assumption 3-p from their paper when one is not considering time dependence (i.e., with $v_t=1$.)
	We begin with left-hand side of Assumption 3-p of [Godichon-Baggioni et al. ,2023](https://arxiv.org/pdf/2205.12549):
	$$
	\begin{align}
	\mathbb{E}\left[\left\| \mathbb{E}\left[ \mathbf{g}_t(\mathbf{w}_t)\right] - \nabla f(\mathbf{w}_t) \right\|^2\right] &\stackrel{(a)}{=} \mathbb{E}[\| \nabla f(\mathbf{w}_t) + \beta_t - \nabla f(\mathbf{w}_t)\|^2] \\\\
	&= ||\, \beta_t ||^2  \\\\
	&\stackrel{(b)}{\leq} P_b\, || \nabla f(\mathbf{w}_t)||^2 + b^2 \\\\
	&= P_b\, || \nabla f(\mathbf{w}_t) - \nabla f(\mathbf{w}^*) ||^2 + b² \\\\
	&\stackrel{(c)}{\leq} L^2 P_b( \| \mathbf{w_t} - \mathbf{w}^*\|^2) + b^2,
	\end{align}
	$$
	where, $(a)$ is by using the biased stochastic gradient structure we proposed in Definition 1 $(\mathbf{g}_t(\mathbf{w}_t) = \nabla f(\mathbf{w}_t) + \beta_t + \zeta_t)$. Inequality $(b)$ is using our Assumption 3. The last inequality $(c)$ is from standard Lipschitz smoothness. To state  [Godichon-Baggioni et al. ,2023](https://arxiv.org/pdf/2205.12549), if there is short-range dependency $v_t$ can decay quicker, and if there are long-range dependencies it decays much slower. In our work, this dependency is removed by setting $v_t = 1$.

## Choosing fixed vs time-varying step size for learning
We select a fixed learning rate as they are simpler to implement, and computationally efficient empirically. Theoretically, it allows for a simplified convergence analysis. Our result still holds if one adopts a time-varying learning rate bounded from above, i.e., $\gamma_t \leq \gamma$  for all $t \geq 0$, where the bound on $\gamma$ is provided in Equation 14. This style of analysis is considered in [Stich et al., 2020](https://www.jmlr.org/papers/volume21/19-748/19-748.pdf).

---

### Author Response · Authors · 2024-08-13
**Addressing other miscellaneous comments**

**Clarifying our Federated Learning setup:** We have clarified our the problem setup in Section 3.1 to read as follows: Since we are optimizing a non-convex (and smooth) function we make no claims about reaching a globally optimal model parameter $\mathbf{w}^*$. Rather, we optimize it iteratively to reach a stationary point.

**Choosing a larger value of noise:** We consider a larger value for noise on the real-world datasets (KDD10, KDD12 and MNIST) and we see that our proposed algorithm performs better than competing baselines. However, as we continue to increase the noise further, this will eventually result in corruption of gradients / model parameters(increased bias) eventually leading to model divergence. We will make this clear in the manusript.

**Covariance matrix in data generation process for linear regression contains correlation:** With the non-diagonal entries being non-zero, correlations between different features arise, implying that some features provide redundant information. Our algorithm is well equipped to handle redundant features and capture the significant features. Therefore, we can expect our algorithm to perform well even in the presence of correlations between features.

**Miscellaneous fixes:** We have fixed the grammatical errors and revised our appendix section to provide fixes mentioned. For our figures, we have increased the figure, label and legend sizes to provide better readability. A revision has been uploaded.

---

### Decision · Action_Editor_ncsy · 2024-10-28

**Recommendation:** Reject

**Comment:**

This work is a revision of an earlier submission to TMLR. Unfortunately, the revisions have not been substantial. Feedback from all reviewers, as well as an additional area chair, was unanimous in stating that the manuscript requires a thorough and careful revision before it can be considered for acceptance.

On the one hand, the direction of this work is very interesting, as it combines biased gradients with compression and local updates. Another noteworthy aspect is the assumption of a power-law distribution for the stochastic gradients, which deviates from standard assumptions. On the other hand, the paper lacks a discussion of the findings that would position this work within the scope of the existing literature. This is an essential step in making the paper compelling for the TMLR community.

This concern was already raised by the first set of reviewers and in my earlier meta-review. Although the paper now discusses Theorem 1, it does not establish any connection to prior results in the literature.

The reviewers noted that one of the revisions inadvertently revealed the authors' identities. However, this did not impact the decision, as most votes had already been collected at that point.

**Audience:**

This work addresses distributed learning over noisy and erroneous wireless channels and designs a method to perform federated learning in this scenario. This topic is of interest to the TMLR community.

**Claims And Evidence:**

This paper extends the FedProx framework by incorporating gradient compression using a count sketch data structure. A convergence analysis is provided, assuming that data transmission is subject to noise and bias.

The work presents a convergence analysis of the proposed method based on assumptions regarding the aggregated received gradient vector on the server. It is demonstrated that, under these assumptions, the method converges to the neighborhood of a stationary point.

However, it is unclear if the proposed method is directly connected to the assumptions made. For example, the proposed count sketch depends on a parameter $k$, yet there is no discussion of how this parameter relates to the quantities defined in Section 5. In this regard, the theoretical results appear somewhat disconnected from the main claim of the paper—namely, that the proposed method is claimed to efficiently handle data heterogeneity.